# 🌟LiveStar: Live Streaming Assistant for Real-World Online Video Understanding

**Zhenyu Yang[1,2], Kairui Zhang[3], Yuhang Hu[1], Bing Wang[1], Shengsheng Qian[1,2],**∗
**Bin Wen[4], Fan Yang[4], Tingting Gao[4], Weiming Dong[1,2], Changsheng Xu[1,2,5]**
[1]Institute of Automation, Chinese Academy of Sciences, [2]University of Chinese Academy of Sciences,
[3]ShanghaiTech University, [4]Kuaishou Technology, [5]Peng Cheng Laboratory
`yangzhenyu2022@ia.ac.cn`

## Abstract

Despite significant progress in Video Large Language Models (Video-LLMs) for offline video understanding, existing online Video-LLMs typically struggle to simultaneously process continuous frame-by-frame inputs and determine optimal response timing, often compromising real-time responsiveness and narrative coherence. To address these limitations, we introduce LiveStar, a pioneering live streaming assistant that achieves always-on proactive responses through adaptive streaming decoding. Specifically, LiveStar incorporates: (1) a training strategy enabling incremental video-language alignment for variable-length video streams, preserving temporal consistency across dynamically evolving frame sequences; (2) a response-silence decoding framework that determines optimal proactive response timing via a single forward pass verification; (3) memory-aware acceleration via peak-end memory compression for online inference on 10+ minute videos, combined with streaming key-value cache to achieve 1.53× faster inference. We also construct an OmniStar dataset, a comprehensive dataset for training and benchmarking that encompasses 15 diverse real-world scenarios and 5 evaluation tasks for online video understanding. Extensive experiments across three benchmarks demonstrate LiveStar's state-of-the-art performance, achieving an average 19.5% improvement in semantic correctness with 18.1% reduced timing difference compared to existing online Video-LLMs, while improving FPS by 12.0% across all five OmniStar tasks. Our model and dataset can be accessed at `https://github.com/yzy-bupt/LiveStar`.

## 1 Introduction

Recent advancements in Large Vision-Language Models (LVLMs) [1–5] have substantially propelled the development of Video Large Language Models (Video-LLMs) [6–10]. These cutting-edge models have been instrumental in pushing the boundaries of multimodal AI systems through innovations in spatial-temporal reasoning [11–13], memory-enhanced processing [14–16], and long-term video understanding [17–19], particularly enhancing capabilities in offline scenarios.

Building on these advances, researchers are increasingly exploring live streaming assistants for online video understanding [20–25], which remain always-on, reason through time, and provide real-time responses. However, existing online Video-LLMs [25–27] struggle to process frame-by-frame inputs and determine response timing, as shown in Fig 1(a). To effectively process continuous video streams and generate timely responses, such models must possess temporal decision-making capabilities—responding at contextually appropriate moments rather than producing repetitive outputs

---

∗Corresponding author: shengsheng.qian@nlpr.ia.ac.cn

39th Conference on Neural Information Processing Systems (NeurIPS 2025).

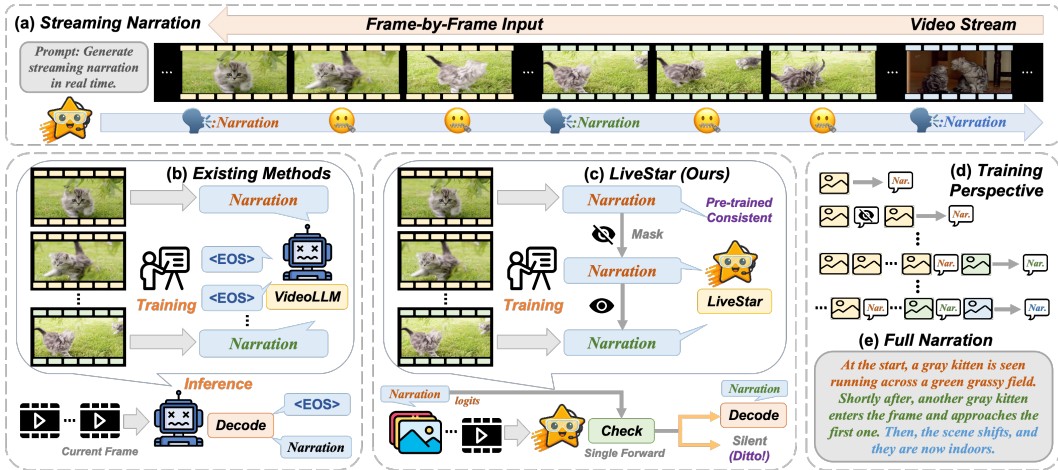

Figure 1: **Illustration of online video understanding.** (a) Taking the RNG task as an example, online video understanding requires Video-LLMs to handle continuous streams and output at appropriate times; (b) Existing methods overly rely on learning the EOS token, leading to poor inference performance; (c)-(e) LiveStar establishes an effective response-silence training and inference framework by SCAM and SVeD without compromising basic video understanding capabilities.

or default "I don't know" responses for each frame. To address this, VideoLLM-online [20] designs a streaming EOS (End-Of-Sequence) prediction mechanism (Fig. 1(b)). It ingests video streams, determines response timing based on user queries, and generates EOS tokens to mark silence intervals. Subsequent advancements like VideoLLM-MoD [21] and LION-FS [22] have further refined this EOS-based framework, achieving improved computational efficiency and response accuracy.

However, current online Video-LLMs exhibit excessive reliance on the EOS token, which leads to four critical issues: (1) **Response-Silence Imbalance**: Frames requiring EOS tokens as output vastly outnumber those generating normal responses [23]. For example, in a 1-minute video at 3 FPS with 5 response intervals, the response-to-silence ratio becomes 1:35. (2) **Consecutive Frame Inconsistency**: As shown in Fig. 1(b), adjacent visually similar frames may produce conflicting outputs—one generating a full narration while the other outputs only the EOS token. This inconsistency disrupts model convergence during fine-tuning. (3) **Pre-training Misalignment**: While pretraining aligns image-text pairs, silent states enforce a mapping from input frames to EOS tokens, contradicting the pretraining objective of meaningful visual-language correspondence. (4) **Vocabulary Confusion**: Since the EOS token is embedded in the vocabulary as a regular token, its frequent appearance in responses pollutes semantic coherence, introducing ambiguity and conflicting with normal outputs. These issues hinder model training effectiveness and degrade base video understanding capabilities, as the forced response-silence mechanism compromises the visual-language alignment. Motivated by this critical gap, we aim to address **Challenge 1**: How to establish an effective response-silence training and inference framework without compromising basic video understanding capabilities?

Moreover, online Video-LLMs face challenges due to limited diversity in training data and evaluation scope, failing to fully encompass the online applications encountered in real-world scenarios. For instance, prominent models [20–22] primarily focus on first-person video analysis through training on Ego4D [28]. Although StreamMind [23] extends its reach to sports scenarios with SoccerNet [29], its scope remains severely constrained. Recent efforts like SVBench [30], OVO-Bench [31], and StreamBench [32] have pioneered online evaluation by assessing models' ability to process videos synchronously during playback. However, these benchmarks focus exclusively on video question answering and lack coverage of diverse real-world tasks. In practice, beyond first-person devices, online scenarios span live streaming platforms, surveillance systems, and cinematic tools, requiring capabilities like real-time interactive editing [33, 34], streaming narration, and temporal grounding in continuous streams. Current evaluation benchmarks present three critical limitations: (1) **Narrow scenario coverage** limited to specific domains, (2) **Single-task evaluation** focused on video question answering, and (3) **Offline assessment** of temporal reasoning and response timing. These shortcomings motivate us to tackle **Challenge 2**: How to construct a comprehensive dataset and

benchmark encompassing diverse real-world scenarios and tasks for holistic evaluation of online video understanding?

To deal with the above challenges, we introduce **LiveStar**, an innovative live streaming assistant that enables online video understanding across diverse scenarios, delivering context-aware responses at semantically relevant moments through adaptive streaming decoding. For **Challenge 1**, we establish a streaming response-silence paradigm featuring two key innovations: (1) a streaming video-language alignment strategy using Streaming Causal Attention Masks (**SCAM**) during training, and (2) a Streaming Verification Decoding (**SVeD**) mechanism for inference. During training, we construct interleaved frame-caption sequences where revealed video frames progressively align with their corresponding captions through causal masked attention constraints (Fig. 1(d)), maintaining consistency with standard multimodal pre-training paradigms. For streaming inference, SVeD employs single-pass verification to determine optimal response timing: it suppresses redundant outputs by comparing current content with previous responses, maintaining narrative continuity through strategic silence. To handle extended video contexts, we implement a peak-end memory compression strategy that prioritizes salient keyframes and final captions, enabling efficient processing of long videos by focusing on semantically rich content while accelerating inference through streaming key-value cache. For **Challenge 2**, we introduce **OmniStar**, a dataset for training and benchmarking that encompasses diverse real-world scenarios and evaluation tasks in online video understanding. The dataset includes carefully annotated real-time narration, streaming QA, and streaming video grounding tasks on 20,137 videos, evaluated through an online setting that better simulates real-world conditions compared to traditional offline settings. Extensive experiments conducted on three benchmarks demonstrate that LiveStar achieves state-of-the-art results for online video understanding.

In summary, our contributions can be summarized as follows:

- We present **LiveStar**, an innovative live streaming assistant for online video understanding, capable of performing real-time comprehension on continuous video streams across diverse real-world scenarios and online tasks, while responding at contextually appropriate moments.

- We design a streaming video-language alignment training strategy using Streaming Causal Attention Masks (**SCAM**). By constructing interleaved frame-caption sequences, we train LiveStar to generate temporally consistent captions incrementally for variable-length streaming video inputs, maintaining narrative coherence across evolving frame prefixes.

- We propose Streaming Verification Decoding (**SVeD**), a response-silence decoding framework that determines optimal response timing via a single forward pass verification. Furthermore, SVeD mimics human-like memory consolidation through a peak-end rule-based strategy, distilling long video contexts by prioritizing salient frames to accelerate processing.

- We introduce **OmniStar**, a comprehensive dataset for training and benchmarking that encompasses 15 diverse real-world scenarios and 5 evaluation tasks in online video understanding. Extensive experiments across three benchmarks demonstrate LiveStar's state-of-the-art performance, achieving an average 19.5% improvement in semantic correctness with 18.1% reduced timing difference compared to existing online Video-LLMs across all five OmniStar tasks.

## 2   Related Work

**Video Large Language Models**   Recent advancements in Large Language Models (LLMs) [35–39] have paved the way for a significant research focus on Video-LLMs [6–10], particularly for enhancing video understanding tasks such as video captioning [40–42], video question answering [43–45] and temporal grounding [46–48] in offline settings. Currently, in addition to the popular closed-source LVLMs such as GPT-4V [49], GPT-4o [37], and Gemini 1.5 Pro [50], an increasing number of open-source Video-LLMs, including LLaVA-NeXT-Video [51], Video-LLaVA [52], and VILA [53], have also demonstrated impressive capabilities in these tasks. Despite these advances, achieving seamless human-computer interaction demands capabilities beyond static video analysis. Specifically, it must process real-time online videos while exhibiting temporal decision-making—responding at contextually appropriate moments and synthesizing histories to maintain coherence [20, 25]. Current Video-LLMs, however, remain limited in handling the dynamic complexities of continuous streaming video processing and often struggle to fully capture the nuances of real-world contexts.

**Video-LLMs for Online Video Understanding**   Recently, there has been growing interest in online video understanding [25, 20], driven by the availability of diverse streaming video sources, including online video platforms [54], live streaming services [55], and wearable camera footage [28]. These resources enable Video-LLMs to learn from real-world contexts, enhancing their ability to interact with streaming content. For instance, VideoStreaming [25] is designed to understand video content in real-time through streaming encoding and memory selection. However, existing models [25–27] struggle to determine response timing and process frame-by-frame inputs. To address this, VideoLLM-online [20] introduces a streaming EOS (End-Of-Sequence) prediction mechanism, which continuously ingests video streams and generates EOS tokens to indicate periods requiring silence. Subsequent models [21, 22] refine this framework, improving computational efficiency and response accuracy. Despite this, excessive reliance on EOS tokens disrupts training and performance. This imbalanced dependency [23] causes most frames to be suppressed into silent outputs, while visually similar adjacent frames produce conflicting predictions (e.g., alternating between captions and silence). Furthermore, it misaligns with pretraining objectives that prioritize visual-language alignment [56, 57], as silent states incentivize trivial mappings rather than meaningful understanding. To bridge this gap, we introduce a live streaming assistant designed to perform real-time understanding of continuous video streams, delivering responses at contextually appropriate moments while maintaining the core capabilities of its pre-trained video-language alignment.

**Online Video Understanding Benchmarks**   The exponential growth of video data has elevated video understanding to a critical field. To rigorously evaluate model capabilities, researchers have established a range of standardized benchmarks [44, 58–68] for offline Video-LLMs. While these benchmarks enable comparative evaluations and drive progress in Video-LLMs, they primarily rely on complete videos or individual clips for offline analysis. Recent efforts like SVBench [30], OVO-Bench [31], and StreamBench [32] have pioneered online evaluation by assessing models' ability to process videos synchronously during playback. However, these benchmarks focus exclusively on video question answering and lack coverage of diverse real-world scenarios. Current Video-LLM evaluations remain constrained, with most models primarily evaluated on the first-person Ego4D dataset [28]. Therefore, we introduce a comprehensive dataset for training and benchmarking that encompasses diverse real-world scenarios and evaluation tasks in online video understanding.

## 3   Method: LiveStar

### 3.1   Training Strategy

We design a streaming training strategy for aligning streaming video and language using Streaming Causal Attention Masks (**SCAM**). By constructing interleaved frame-caption sequences, we train LiveStar to incrementally generate temporally consistent captions for streaming video inputs of varying lengths, maintaining narrative coherence across evolving prefixes as the frames progress.

**Streaming Video-Language Alignment**   Current Video-LLMs primarily rely on static vision-language foundation models [1, 2, 69] pre-trained on image-text pairs, exhibiting limited adaptation to streaming video-text relationships [10]. These models typically optimize the objective by image/video-text pair alignment:

$$\max P([Txt_i] \mid [Img_i]/[Vid_i]), \tag{1}$$

which fails to address two critical challenges in online video understanding: (1) incremental processing of streaming frames, and (2) dynamic alignment between evolving visual contexts and linguistic outputs. To address them, we propose multi-turn, frame-by-frame instruction tuning that bridges streaming video inputs with language generation through the reformulated objective:

$$\max P([Txt^k] \mid [Ctx^{<t_i}], [Frm^{t_i}]), \ \forall t_i \in C_k, \tag{2}$$

where $[Frm^{t_i}]$ denotes the video frame at timestamp $t_i$, $[Ctx^{<t_i}]$ represents the accumulated multi-modal context, and $C_k = \{t_i\}_{i=m}^n$ indicates a **semantic clip** (frames from $t_m$ to $t_n$ within the same scene/event/action) sharing semantic text $[Txt^k]$. This addresses the pretraining misalignment in existing EOS-based models [20–23] that enforce end-of-sequence token prediction (i.e., $\max P(\text{EOS} \mid [Ctx^{<t_i}], [Frm^{t_i}])$) across non-response frames. Unlike these models, LiveStar achieves streaming consistency through incremental context integration over variable-length video streams. Our method maintains narrative coherence by dynamically updating frame prefixes while preserving evolving visual contexts, establishing *a new paradigm* for streaming video understanding.

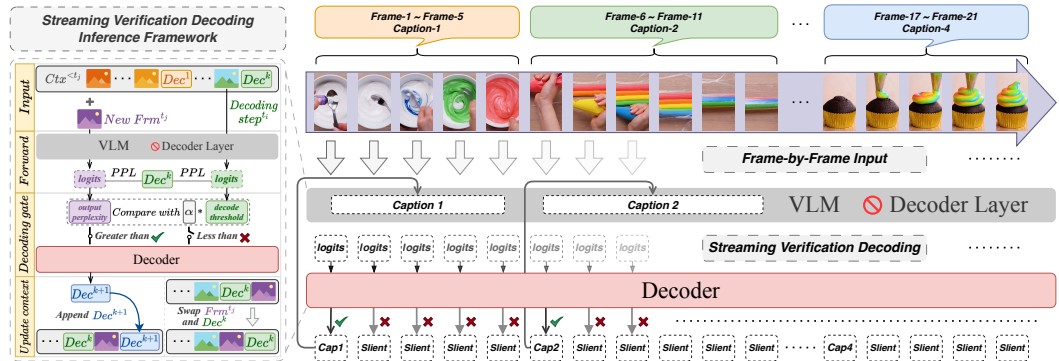

Figure 2: **Overview of the streaming verification decoding (SVeD) inference framework:** A dynamic response-silence decoding framework designed to determine optimal response timing for online video understanding.

**Interleaved Frame-Caption Sequences** To enable multi-turn, frame-by-frame instruction tuning, we construct interleaved frame-caption sequences using a chat-inspired format, which allows LiveStar to incrementally ingest visual inputs while preserving temporal awareness for accurate scene transition detection. Each conversational turn includes: (1) a frame $[Frm^{t_i}]$ (at timestep $t_i$) and (2) a corresponding caption $[Cap^k]$ (for the $k$-th semantic clip). Consecutive frames from the same semantic clip share captions with identical semantics. To mitigate overfitting from repeated caption exposure, we randomly sample a caption $[Cap_j^k]$ from a pool of $M$ paraphrased captions.

**Streaming Causal Attention Masks** To train LiveStar autoregressively on interleaved frame-caption sequences, we address three challenges: (1) Preventing leakage: Already-generated captions within the current semantic clip must be masked when generating the current caption to prevent trivial copying, as they are identical in content. (2) Token-specific context: During autoregression, the model must retain visibility into previously predicted tokens for the current caption to maintain coherence. (3) Scene transition signaling: The last caption of each semantic clip must persist across subsequent frames to explicitly demarcate semantic boundaries. To address these, we propose Streaming Causal Attention Masks (SCAM) (see Fig. 3 for the mask matrix), replacing standard causal attention masks. Formally, when generating the caption for frame $Frm^{t_i}$ in the clip $C_k$, the model is constrained to attend only to all video frames from the preceding clips $\{C_1, C_2, \ldots, C_{k-1}\}$ and the last captions of those clips. The final optimization objective is as follows:

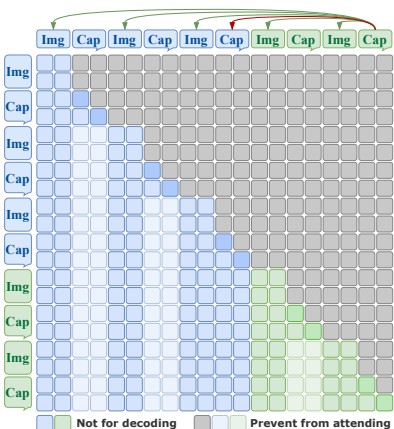

Figure 3: Mask matrix of SCAM.

$$\max P([Cap_j^k] \mid [Ctx^{<t_i}\{Mask^{\leq t_i}\}], [Frm^{t_i}]), \ \forall t_i \in C_k. \tag{3}$$

Here, the mask matrix $Mask^{\leq t_i}$ is designed to block attention to all non-terminal caption tokens from semantic clips $\{C_1, C_2, \ldots, C_k\}$ prior to timestep $t_i$, as illustrated in Fig. 1(d).

## 3.2 Inference Framework

While LiveStar can generate real-time captions at each timestep, determining when to update the response while maintaining content coherence and avoiding redundant outputs remains a critical challenge. To address this, we propose a dynamic response-silence decoding (SVeD) framework (see Fig. 2) and employ memory-aware techniques for acceleration.

**Streaming Verification Decoding** We present SVeD, a dynamic response-silence decoding framework designed to determine optimal response timing for online video understanding. It intro-

duces a *decoding gate* that selectively triggers caption generation based on a streaming verification mechanism. At each triggered decoding step $t_i$, we compute the perplexity of the generated caption $[Dec]$ as: $\text{PPL}^{t_i}([Dec]) = \sqrt[N]{1/P([Dec] \mid [Ctx^{<t_i}], [Frm^{t_i}])}$, where $N$ represents the token count of $[Dec]$, and $P(\cdot)$ is the autoregressive probability derived from token logits. For each incoming frame $[Frm^{t_j}]$, SVeD performs a single forward pass to verify the latest caption's validity by recomputing $\text{PPL}^{t_j}([Dec])$. If $\text{PPL}^{t_j}([Dec]) > \alpha \cdot \text{PPL}^{t_i}([Dec])$ (where $\alpha$ is a tunable scaling factor), the gate activates decoding at $t_j$, generating an updated caption. Otherwise, move $[Dec]$ to the end of $[Ctx]$ without decoding, preserving temporal coherence while minimizing latency. Under the same model architecture, SVeD achieves faster inference than decoding an EOS token to indicate a silent response. This lightweight verification step ensures adaptive response timing, balancing accuracy and efficiency. For details, refer to Alg. 1.

**Peak-End Memory Compression** Modern streaming videos often span hours with high frame rates, posing computational challenges for long-term understanding. Inspired by the Peak-End cognitive rule [70–72]—where human memory retention prioritizes salient moments (keyframes) and recent experiences (summaries)—we propose a memory compression framework tailored for 10+ minute video analysis at 3 fps. Our method leverages two critical signals: (1) *Keyframe detection*: We have computed the perplexity $\text{PPL}^{t_i}([Dec])$ for each frame, where lower values indicate higher semantic importance. (2) *Temporal summarization*: The final frame's caption of each semantic clip encapsulates event semantics. To optimize memory, we probabilistically prune frames older than a window $W$, with deletion likelihood proportional to both the relative PPL within its semantic clip and elapsed time. Experiments demonstrate that it achieves optimal semantic accuracy (SemCor) and minimal timing difference (TimDiff) compared to Uniform Dropout and FIFO Forgetting, as shown in Tab. 4.

---

**Algorithm 1:** Streaming Verification Decoding

**Input:** Video frame stream $\{[Frm^t]\}_{t=1}^T$
**Output:** Dynamically generated caption $[Dec]$
Initialize $[Dec], [Ctx] \leftarrow \emptyset$
Initialize reference timestamp $t_i \leftarrow 0$
**for** *each incoming frame* $[Frm^{t_j}]$ **do**
    Append $[Frm^{t_j}]$ to $[Ctx]$
    **if** $[Dec] \neq \emptyset$ **then**
        Compute verification perplexity:
        $\text{PPL}^{t_j}([Dec]) = \sqrt[N]{\dfrac{1}{P([Dec] \mid [Ctx^{\leq t_j}])}}$
        **if** $PPL^{t_j}([Dec]) > \alpha PPL^{t_i}([Dec])$ **then**
            Activate decoding: Generate new tokens $[Dec]_{\text{new}}$ using $[Ctx^{\leq t_j}]$
            Update $[Dec] \leftarrow [Dec]_{\text{new}}$
            Append $[Dec]$ to $[Ctx]$
            $t_i \leftarrow t_j$
        **else**
            Swap the last two elements in $[Ctx]$
            `// Move` $[Dec]$ `to the end`
    **else**
        Perform initial decoding to generate $[Dec]$
        Append $[Dec]$ to $[Ctx]$
        $t_i \leftarrow t_j$

---

**Streaming Key-Value Cache.** We implement an effective streaming key-value (KV) cache mechanism for multi-turn conversational interactions. We introduce a dual-level caching architecture that maintains both intra-dialogue KV caches for frame-level processing and inter-dialogue streaming caches for long-context preservation across conversations. This eliminates redundant recomputation of historical context representations (keys and values) during new video frame processing while ensuring low-latency inference. It specifically addresses two critical challenges: (1) Cache sequence integrity under swap operations in our SVeD module, and (2) Dynamic length adaptation through Peak-End Memory Compression that strategically prunes redundant tokens while preserving temporal coherence. Experiments (see Tab. 4) demonstrate that it achieves 1.53× faster inference than methods without KV caching while incurring negligible performance loss on video streams.

## 4 Dataset: OmniStar

**Datasets Statistics.** We introduce OmniStar, a comprehensive online video understanding dataset featuring 15 diverse real-world scenarios (46 specific categories) and 5 online evaluation tasks. The dataset comprises 20,137 expert-annotated video streams with temporally dense annotations, rigorously split into 19,137 training and 1,000 evaluation instances (200 per task) without overlap. It establishes five synergistic evaluation tasks through a unified streaming protocol: (1) Real-time narration generation [RNG], (2) Online temporal grounding [OTG], (3) Frame-level dense QA [FDQ], (4) Contextual online QA [COQ], and (5) Multi-turn interactive QA [MIQ]. More detailed online tasks are described in detail in App. D. The 15 scenarios include: Travel & Events, Sports, Pets & Animals, Music, Autos & Vehicles, Film & Animation, Nonprofits & Activism, Science &

Technology, Education, Howto & Style, News & Politics, Entertainment, Comedy, People & Blogs, and Gaming, each subdivided into 2-4 fine-grained categories using YouTube's native annotation system.

**Temporally Dense Annotation.** In contrast to prior online video understanding benchmarks, which are typically limited to single question-answering task and lack coverage of real-world scenarios [30–32], OmniStar adopts a semi-automated, temporally dense annotation pipeline to obtain denser labeled data. Every video is annotated with multiple caption segments, where each segment is precisely aligned with its corresponding start and end timestamps. Instead of providing disjointed descriptions, OmniStar's annotations are constructed as narrative-consistent caption sequences that collectively form coherent storylines (e.g., *"First, the woman gathers the ingredients..."* → *"Then, she mixes them in a large bowl..."* → *"Finally, she places the dish in the oven..."*), thereby facilitating the RNG task by effectively modeling long-range narrative flow.

Building on this foundation, OmniStar introduces two forms of streaming QA supervision: (1) temporally distributed multi-turn QA chains (Task: MIQ, COQ), where earlier and later questions and answers are semantically linked, and (2) frame-level streaming QA (Task: FDQ), where a standing query corresponding to **time-varying answers** (e.g., *Q: "What is the person showing?"* → *"A pen."* → *"A book."* → *"A smartphone."*). Both supervision paradigms are designed to move beyond static, isolated QA settings by encouraging models to maintain temporal awareness, integrate past context, and update answers according to the video streams. Case examples are listed in Tabs. 7, 8, and 9.

# 5 Experiments

## 5.1 Experimental Setup

**Datasets.** We validate LiveStar's effectiveness across offline and online settings, including Ego4D [28] (egocentric narration stream), SVBench [30] (streaming QA) and OmniStar (5 real-world online tasks). We build the Ego4D Narration Stream benchmark using dense timestamped narrations, following VideoLLM-online [20]. We leverage SVBench's evaluation set for dialogue and streaming QA. For real-world online evaluation, we benchmark Video-LLMs on OmniStar through five online tasks, with real-time narration generation assessed in both offline and online settings. We design a two-phase progressive training paradigm for LiveStar, detailed in App. A.2.

**Implementation Details.** We conducted full fine-tuning of LiveStar on 8× NVIDIA A800 GPUs. Building upon InternVideo2.5 [10], the model consists of a vision encoder (InternViT [1]), an MLP projector, and a large language model (InternLM2.5-7B [73]). For the vision encoder, we employ InternViT to extract video frame embeddings at a rate of 1-4 FPS, with each frame represented by 16 tokens. To optimize efficiency in online video understanding, we fine-tuned the model using a static resolution strategy, enabling it to process multi-minute video content within an 8K-token context window. The extracted frame embeddings are then fed into an MLP projector to generate frame tokens, following the approach used in LLaVA-1.5 [74]. These frame tokens are interleaved with language tokens and input into the large language model, InternLM2.5. Additional details about model architecture and model training can be found in App. A.1.

**Parameter Settings.** During training, we conducted full fine-tuning of LiveStar using a total of 83K data, which includes the OmniStar training set. We trained the models for 1 epoch with a learning rate of $4 \times 10^{-5}$ using the AdamW optimizer ($\beta_1 = 0.9$, $\beta_2 = 0.999$, weight decay = 0.05), a per-device batch size of 1, and gradient accumulation over 4 steps to achieve an effective batch size of 32. We adopted cosine learning rate scheduling with a warmup ratio of 0.03. Input frames were uniformly resized to $448 \times 448$, with a patch downsampling ratio of 0.5. The vision encoder was frozen during training, while the MLP projector and language model components were fully updated. Each training sequence contains up to 8192 tokens, consisting of interleaved frame and language tokens following the InternVL2.5 conversational template. We optimized the model using the standard autoregressive cross-entropy loss computed over the language tokens, where loss was computed only on assistant response tokens, and inter-frame language segments were excluded via our SCAM strategy. For inference, the tunable scaling factor in SVeD was set to 1.03 by default, the prune window W in peak-end memory compression was set to 40 frames, and the size of the paraphrased caption pool of streaming video-language alignment was set to M=1 by default for better temporal alignment.

Table 1: Evaluation results of various models on OmniStar-RNG in offline and online evaluation. PPL is a commonly used metric for narration, but due to the different vocabulary sizes of LLMs, it is only for reference. "-" indicates the inability to perform the required test.

| Method | Offline Evaluation (Fixed decoding) | | | | Online Evaluation (Response-silence decoding) | | | | |
|---|---|---|---|---|---|---|---|---|---|
| | PPL↓ | TokAcc↑ | SemCor↑ | SumFluen↑ | TimDiff↓ | TimRedun↓ | TimCover↑ | SemCor↑ | SumFluen↑ |
| Human | - | - | 6.73 | 7.17 | 1.08 | 1.24 | 0.84 | 6.09 | 6.81 |
| **Offline Video-LLMs/LVLMs** | | | | | | | | | |
| GPT-4V | - | - | 4.97 | 5.37 | - | - | - | - | - |
| GPT-4o | - | - | 5.03 | 5.45 | - | - | - | - | - |
| LLaVA-Video | 12.42 | 0.53 | 3.40 | 2.88 | - | - | - | - | - |
| InternVideo2.5 | 6.91 | 0.56 | 4.32 | 3.61 | - | - | - | - | - |
| InternVL2.5 | 9.81 | 0.51 | 3.40 | 2.94 | - | - | - | - | - |
| MiniCPM-V 2.6 | 9.46 | 0.57 | 4.34 | 4.13 | - | - | - | - | - |
| Qwen2.5-VL | 13.80 | 0.59 | 4.42 | 4.24 | - | - | - | - | - |
| **Online Assistants** | | | | | | | | | |
| VideoLLM-online | 9.73 | 0.49 | 3.01 | 0.69 | 2.67 | 2.15 | 0.80 | 1.68 | 0.59 |
| VideoLLM-MoD | 9.93 | 0.48 | 2.89 | 0.65 | 2.54 | 2.49 | **0.90** | 1.66 | 0.55 |
| MMDuet | 5.69 | 0.60 | 4.29 | 3.40 | 2.32 | **0.62** | 0.51 | 1.93 | 2.69 |
| LiveStar (Ours) | **5.14** | **0.62** | **4.62** | **4.55** | **1.91** | 0.95 | 0.71 | **3.19** | **4.25** |

## 5.2 Online Experiments

To simulate real-world scenarios of online video understanding, we evaluated Video-LLMs during the inference phase using the OmniStar test set. Unlike the Ego4D Narration Stream benchmark, which is offline due to the absence of scoring for inference results and fixed decoding points (as is the case with SVBench), our OmniStar allows Video-LLMs to determine the response-silence timing across five tasks in an online setting. This evaluation not only assesses the accuracy of the output timing but also examines whether the output quality aligns with the ground truth.

**Evaluation Metrics.** We use the following metrics to evaluate the model as an online video assistant:

- **TimDiff (Timing Difference)** quantifies the absolute time difference between model responses and ground-truth annotations (missing outputs are penalized with full scene duration; multiple outputs are penalized by cumulative time), evaluating real-time response precision. Lower values indicate better synchronization with visual events.

- **TimRedun (Timing Redundancy)** measures redundancy through the average number of unnecessary responses per scene, where a perfect single response could fully capture the scene content.

- **TimCover (Timing Coverage)** computes the average coverage of responses across all scenes, where a scene is scored 1 if it contains at least one valid response and 0 otherwise.

- **SemCor (Semantic Correctness)** scores semantic alignment between responses and ground truth using GPT-4o across three axes: (1) Semantic Accuracy, (2) Language Quality, and (3) Information Completeness. The final SemCor averages all dimensions (0–10 each). See App. E for details.

- **SumFluen (Summarize Fluency)** evaluates holistic fluency of concatenated responses against aggregated ground truth using GPT-4o, with scores for (1) Writing Logicality, (2) Language Fluency, (3) Writing Conciseness, (4) Semantic Consistency, and (5) Narrative Completeness.

**Results.** The right side of Tab. 1 presents the results of various models on OmniStar-RNG in online evaluation (under each model's own response-silence strategy), indicating that LiveStar achieves lower response latency and higher semantic precision. Notably, VideoLLM-online and VideoLLM-MoD output on nearly every frame, leading to the highest TimCover, but this aggressive strategy also results in poorer

Table 2: Evaluation results of online Video-LLMs on OmniStar. "-" in OTG means no generation needed for scoring; "-" in COQ and MIQ indicates real-time QA. Note that VideoLLM-online did not achieve the reported 10 FPS.

| Method | Online Evaluation (SemCor↑/TimDiff↓) | | | | | FPS↑ (5min) |
|---|---|---|---|---|---|---|
| | RNG | OTG | FDQ | COQ | MIQ | |
| Human | 6.09/1.08 | -/1.81 | 9.12/1.01 | 7.96/- | 7.83/- | - |
| VideoLLM-online | 1.68/2.67 | -/9.69 | 2.35/2.15 | 4.01/- | 3.83/- | 3.37 |
| VideoLLM-MoD | 1.66/2.54 | -/9.83 | 2.11/2.23 | 3.99/- | 3.75/- | 3.41 |
| MMDuet | 1.63/2.32 | -/4.42 | 4.78/2.65 | 5.71/- | 5.62/- | 0.91 |
| LiveStar (Ours) | **3.19/1.91** | **-/3.57** | **6.44/1.80** | **5.85/-** | **5.78/-** | **3.82** |

performance on other metrics. In contrast, MMDuet produces much sparser outputs, resulting in the lowest TimRedun, but at the cost of lower scores on other metrics. These deficiencies are clearly

demonstrated in Tab. 2, where LiveStar consistently outperforms all other online Video-LLMs across 5 tasks of the OmniStar benchmark while maintaining the fastest inference speed-with a 19.5% increase SemCor score, an 18.1% reduction TimDiff value, and a 12.0% increase in FPS compared to the second-best model.

## 5.3 Offline Experiments

Unlike online evaluation where models *autonomously determine response timing*, our offline evaluation on Ego4D, OmniStar-RNG, and SVBench pre-specifies decoding timing. For fairness, we strictly adopt the evaluation protocol and metrics from previous works [20–22] despite their non-generative constraints (i.e., ground-truth verification only). On OmniStar, we extend beyond standard perplexity and token accuracy verification for ground truth to implement full model generation with online scoring – a critical advancement enabling comprehensive online capability assessment.

**Evaluation Metrics.** For rigorous benchmarking, we adopt established offline metrics: PPL, TokAcc (Token Accuracy, previously referred to as LM-Correctness), TimeDiff, and Fluency from prior works [20–22]. Additionally, we implement online generation evaluation with offline prescribed timing, measured through SemCor and SumFluen metrics, both scored based on GPT-4o.

Table 3: Evaluation results of online Video-LLMs on Ego4D Narration Stream benchmark in offline settings. "-" indicates the model lacks EOS tokens for Fluency.

| Method | PPL↓ | TimeDiff↓ | Fluency↑ | TokAcc↑ |
|---|---|---|---|---|
| VideoLLM-online | 2.43 | 2.04 | 45.1% | 48.1% |
| VideoLLM-MoD | 2.41 | 2.04 | 45.2% | 48.9% |
| LION-FS | 2.09 | 2.15 | **46.1%** | 52.4% |
| MMDuet | 4.51 | 1.97 | - | 39.3% |
| LiveStar (Ours) | **1.97** | **1.76** | - | **61.1%** |

**Results.** The left side of Tab. 1 presents the results of various models on OmniStar-RNG in offline evaluation (given decoding timing conditions), showing that LiveStar surpasses all other online assistants and open-source offline Video-LLMs/LVLMs across all metrics, though there remains a performance gap compared to human performance and GPT-4V/4o. Tab. 3 demonstrates that LiveStar comprehensively outperforms all other online assistants on the Ego4D Narration benchmark, particularly achieving an 8.7% higher TokAcc than the second-best LION-FS. Experimental results and analyses for dialogue and streaming evaluations on SVBench can be found in App. C.1.

## 5.4 Ablation Study

**Impact of Response-Silence Threshold.** LiveStar employs a dynamic response-silence decoding framework governed by an adaptive threshold mechanism. The decoding threshold is defined as $\alpha \cdot \text{PPL}^{t_i}([Dec])$, where $\alpha \geq 1$ is a tunable scaling factor applied to the timestep-specific perplexity. To rigorously evaluate the parameter sensitivity of $\alpha$, we conducted an empirical analysis over the interval $[1.0, 1.1]$ using the OmniStar-RNG benchmark. Our online evaluation (see Fig. 4) demonstrates $\alpha$'s critical role in balancing timing difference, timing redundancy, and timing coverage. Controlled tests revealed optimal performance within the narrow range $\alpha = 1.02$–$1.04$, with $\alpha = 1.03$ selected as the final configuration.

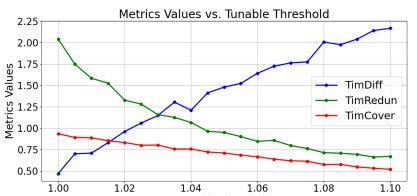

Figure 4: Ablation study on the impact of response-silence threshold.

**Strategy for Inference Acceleration.** Tab. 4 presents ablation studies on memory dropout and KV cache strategies for inference acceleration. For memory dropout mechanisms: (1) Uniform dropout causes a 4.70% SemCor degradation due to critical recent frames being discarded; (2) FIFO forgetting impairs temporal reasoning by discarding historical event captions, increasing TimDiff by 9.42% with 3.76% SemCor reduction; (3) Our Peak-End Memory Compression preserves seman-

Table 4: Ablation study on strategies for inference acceleration on OmniStar-RNG.

| Dropout | KV Cache | SemCor↑ | TimDiff↓ | FPS↑ |
|---|---|---|---|---|
| Uniform | Both | 3.04 | 2.01 | 3.77 |
| FIFO | | 3.07 | 2.09 | 3.91 |
| Peak-End | Neither | 3.19 | 1.95 | 2.50 |
| | w/o Inter-Dialog | 3.17 | 1.87 | 2.92 |
| | Both | 3.19 | 1.91 | 3.82 |

tic clip summaries and keyframes through precomputed PPL dropout of non-critical frames, achieving optimal SemCor and minimal TimDiff. Regarding KV cache: While removing both inter/intra-dialog caching or intra-dialog caching shows negligible performance loss, adopting both strategies boosts

FPS by 1.53×/1.31× respectively under 5-minute-videos inference by eliminating redundant recomputation of historical key-value representations during new frame processing, enabling low-latency inference without quality degradation.

**Size of Paraphrased Caption Pool.** To mitigate overfitting induced by repeated caption exposure during training, we employ a stochastic sampling approach that selects captions $[Cap_j^k]$ from a paraphrased caption pool of size $M$. Experiments reveal a critical trade-off between semantic richness and temporal consistency: Increasing $M$ enhances semantic correctness (SemCor), with $M = 3$ achieving a +1.57% improvement over $M = 1$, but simultaneously degrades temporal alignment

Table 5: Ablation study on the size of the caption pool.

| Pool Size | SemCor↑ | TimDiff↓ |
|---|---|---|
| M=1 | 3.19 | **1.91** |
| M=2 | 3.23 | 1.98 |
| M=3 | **3.24** | 1.97 |

(TimDiff) by +3.14%. While larger pools better capture semantic diversity, the incurred temporal inconsistency becomes prohibitive for real-time applications. Given the heightened sensitivity of temporal alignment in online tasks, we adopt $M = 1$ as the default configuration. This framework allows task-aware adaptation, enabling practitioners to optimize the pool size according to their specific robustness-latency requirements.

# 6 Case Study

We conduct qualitative comparisons across multiple tasks, including RNG, MIQ, COQ, and FDQ, between our LiveStar model and existing representative online video understanding models, VideoLLM-online and MMDuet [24], with details shown in Figs. 5, 6 and 7. These cases demonstrate that VideoLLM-online and MMDuet often suffer from limited contextual understanding, hallucinations, and insufficient fine-grained recognition, while LiveStar consistently provides more accurate, grounded, and timely responses. These comparisons collectively highlight the advantages of LiveStar and reveal the limitations of current LVLMs in handling fine-grained and context-sensitive video question answering.

# 7 Conclusion

This paper introduces LiveStar, an innovative live streaming assistant that pioneers always-on proactive responsiveness through adaptive streaming decoding. We establish a streaming response-silence paradigm featuring two technical contributions: (1) a streaming video-language alignment framework utilizing Streaming Causal Attention Masks (SCAM) during model training, and (2) a Streaming Verification Decoding (SVeD) mechanism for real-time inference optimization. Through extensive experiments across three benchmark datasets, LiveStar demonstrates state-of-the-art performance in online video understanding while achieving practical deployment capabilities of processing 10+ minute video streams at 3 FPS. This paper significantly advances the field by introducing a novel response-silence paradigm and establishing a comprehensive benchmark, thereby stimulating future research in developing advanced models for complex online video understanding tasks.

**Limitations.** Despite the superior performance of our proposed LiveStar on several benchmarks for online video understanding, there remain some limitations: First, to improve inference efficiency, we compress each video frame into 16 visual tokens. While this compact representation significantly reduces computational cost, it inevitably compromises the model's ability to capture fine-grained visual details. As a result, the model may miss critical cues in scenarios involving subtle motion changes or complex scene dynamics. Second, the current version of LiveStar only supports vision-text modalities and does not incorporate audio information. This limitation constrains the model's capacity to fully leverage multimodal cues in video understanding tasks. In future work, we aim to extend the model to include audio modalities for more comprehensive multimodal reasoning.

# Acknowledgment

We would like to present our appreciation to the anonymous reviewers and ACs for their constructive suggestions. This work is supported by the National Key Research and Development Program of China (No.2023YFC3310700), the Beijing Natural Science Foundation (JQ23018, L252032), and the National Natural Science Foundation of China (No.62036012, 62276257).

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

# Appendix

## A  Implementation Details

### A.1  Model Architecture

Inspired by InternVideo2.5-Chat-8B [10], the model consists of a vision encoder (InternViT [1]), an MLP projector, and a large language model (InternLM2.5-7B [73]). For the vision encoder, we employ InternViT, a model pre-trained on a hybrid dataset combining image captioning and OCR-specific data, to extract video frame embeddings at a rate of 1-4 FPS, with each frame represented by 16 tokens. To optimize efficiency in online video understanding, we fine-tuned the model using a static resolution strategy, enabling it to process multi-minute video content within an 8K-token context window. The extracted frame embeddings are then fed into an MLP projector to generate frame tokens, following the approach used in LLaVA-1.5 [74]. These frame tokens are interleaved with language tokens and input into the large language model, InternLM2.5.

### A.2  Training Datasets

Our proposed LiveStar builds upon the pre-trained InternVideo2.5 foundation model [10], which demonstrates robust capabilities in offline video comprehension through spatial-temporal reasoning and long-term temporal modeling, achieved through large-scale pretraining on 3.7M video-text pairs. To bridge the gap between static video understanding and dynamic streaming perception, we design a two-phase progressive training paradigm, implementing streaming video-language alignment through Streaming Causal Attention Masks (SCAM) as described in Sec. 3.1, to enable effective adaptation to online video understanding scenarios.

**Phase I: Temporal Alignment Pretraining.** We conduct dense video-language alignment using 63K carefully curated video segments from carefully selected high-quality *subsets* of three complementary datasets: ActivityNet Captions [59] (9K selected from 20K raw samples), Shot2Story [75] (33K selected from 43K raw samples), Ego4D Narration Stream [28] (20K selected from 113K raw samples), and MVBench [44] (1K selected from 4K raw samples). This stage focuses on establishing frame-level semantic correspondences and enhancing the model's capability to process streaming visual input through SCAM.

**Phase II: Multi-Task Online Adaptation.** We employ the OmniStar benchmark's 20K training samples across five distinct online video understanding tasks: (1) Real-time narration generation, (2) Online temporal grounding, (3) Frame-level dense QA, (4) Contextual online QA, and (5) Multi-turn interactive QA. Through task-specific adapters, we achieve simultaneous multi-objective alignment that preserves general video understanding capabilities while specializing for streaming scenarios.

This hierarchical training strategy demonstrates two key advantages: 1) Effective knowledge transfer from static to streaming domains through progressive alignment, and 2) Efficient utilization of limited annotated data (83K total samples) via our SCAM-enhanced streaming video-language alignment framework.

## B  Baselines

In this paper, we present a comparative analysis of online and offline Video-LLMs/LVLMs, evaluating their capabilities in two critical aspects of video understanding: (1) determining optimal response timing and (2) processing continuous video streams. Our study focuses on benchmarking their performance in dynamic, real-world scenarios. We categorize baseline models into two groups based on their temporal processing paradigm. **Online models** are evaluated for their ability to incrementally analyze streaming video frames with minimal timing difference, with the key feature being their ability to autonomously decide when to output. In contrast, **offline models** process complete video sequences as static inputs or can handle streaming video but cannot autonomously decide when to output. This dichotomy allows us to systematically compare temporal reasoning strategies and online video understanding capabilities across different architectures.

**Online Video-LLMs:**

- VideoLLM-online [20]: a streaming-oriented architecture employs continuous key-value cache to enable low-latency incremental processing of video streams. It designs a streaming EOS (End-Of-Sequence) prediction mechanism to determine response timing based on user queries and generates EOS tokens to mark silence intervals.

- VideoLLM-MoD (reproduced) [21]: a modified variant of VideoLLM-online utilizing mixture-of-depths token selection to balance computational efficiency and temporal reasoning accuracy in streaming video analysis. Since the code is not open-sourced, we reproduced it based on VideoLLM-online.

- MMDuet [24]: a multimodal dialogue-oriented model featuring dual-stream encoders for synchronized audio-visual processing is specifically designed for interactive video applications. Unlike VideoLLM-online and VideoLLM-MoD, this model uses a classification head to decide whether to proceed with decoding and output.

- LION-FS (paper-reported results) [22]: a dual-path video-language framework that resolves the efficacy-efficiency trade-off in online video assistance through dynamic routing-based real-time response determination and multi-granular keyframe augmentation with spatial-interaction modeling. Since it is not open-sourced, we used the reported results from the original paper for comparison on the Ego4D Narration Stream benchmark.

These models use their respective response-silence decoding strategies to effectively process continuous video streams and generate timely responses. They possess temporal decision-making capabilities, responding at contextually appropriate moments, processing frame-by-frame inputs, and determining response timing.

**Offline Video-LLMs/LVLMs:**

- GPT-4V & GPT-4o [76]: both models process real-time video streams alongside synchronized audio, text (e.g., subtitles), and visual frames, enabling holistic comprehension of dynamic scenes. Their autoregressive architecture supports sequential processing of video frames, capturing temporal dependencies critical for tasks like action recognition, event prediction, or summarizing long-form content.

- LLaVA-NeXT-Video [77]: an open-source chatbot advances video understanding through zero-shot AnyRes representation and linear scaling for long-context processing.

- InternVideo2.5 [10] & InternVL2.5 [1]: the first open-source MLLMs to achieve over 70% on the MMMU benchmark, matching the performance of leading closed-source commercial models like GPT-4o.

- MiniCPM-V 2.6 [3]: the latest and most capable model in the MiniCPM-V series.

- Qwen2.5-VL [78]: the flagship multimodal model in the Qwen series, advances visual-language intelligence through a native dynamic-resolution ViT with Window Attention, enabling efficient high-fidelity processing of variable-sized images and hour-long videos with second-level temporal localization.

*Implementation Notes:* Due to the lack of open-source code for VideoLLM-MoD, we reproduced it based on the VideoLLM-online framework. For LION-FS, which is also not open-sourced, we utilized the results reported in the original paper for comparison on the Ego4D Narration Stream benchmark. StreamMind was excluded from our comparisons due to the absence of open-source datasets and model weights.

# C  Additional Experiments

## C.1  Dialogue and Streaming Evaluations on SVBench

We benchmarked open-source Video-LLMs/LVLMs and closed-source LVLMs aligned with OmniStar on SVBench. The dialogue and streaming evaluation results on SVBench, outlined in Table 6, provide a comprehensive comparison among various Video-LLMs. We have published the evaluation results on the official leaderboard website[2]. Notably, closed-source models such as GPT-4o and GPT-4V attain significantly higher scores across all metrics, with GPT-4o achieving an Overall Score (OS) of 62.57 in dialogue evaluation and 59.97 in streaming evaluation. LiveStar's zero-shot performance on SVBench in dialogue and streaming evaluations outperforms LLaVA-NeXT-Video, InternVL2.5, and InternVideo2.5, but underperforms MiniCPM-V 2.6 and Qwen2.5-VL. However, after fine-tuning on the SVBench training set, LiveStar achieves an average improvement of 15.37% across metrics, surpassing all open-source Video-LLMs/LVLMs and closing the gap with GPT-4V. This demonstrates LiveStar's robust online video understanding capabilities.

Table 6: Evaluation results of various models on SVBench in dialogue and streaming evaluation. † denotes fine-tuning on the SVBench training set.

| Model | Dialogue Evaluation | | | | | | Streaming Evaluation | | | | | | AVG |
| --- | --- | --- | --- | --- | --- | --- | --- | --- | --- | --- | --- | --- | --- |
| | SA | CC | LC | TU | IC | OS | SA | CC | LC | TU | IC | OS | |
| **Closed-source LVLMs** | | | | | | | | | | | | | |
| GPT-4V | 56.03 | 62.61 | 69.09 | 65.36 | 53.73 | 60.30 | 56.37 | 61.41 | 65.80 | 59.18 | 57.16 | 57.93 | 59.12 |
| GPT-4o | 58.26 | 64.76 | 70.75 | 67.68 | 55.82 | 62.57 | 57.99 | 63.52 | 67.72 | 60.18 | 59.25 | 59.97 | 61.27 |
| **Open-source Video-LLMs/LVLMs** | | | | | | | | | | | | | |
| LLaVA-NeXT-Video | 37.71 | 44.59 | 52.05 | 41.80 | 36.58 | 41.40 | 34.29 | 39.68 | 47.65 | 35.33 | 36.68 | 36.12 | 38.76 |
| InternVL2.5 | 43.73 | 50.70 | 56.61 | 55.03 | 43.46 | 48.73 | 40.44 | 48.34 | 52.84 | 46.93 | 45.10 | 45.04 | 46.89 |
| InternVideo2.5 | 46.83 | 53.48 | 58.22 | 58.91 | 47.02 | 51.73 | 41.76 | 49.72 | 53.25 | 48.44 | 47.10 | 46.58 | 49.16 |
| MiniCPM-V 2.6 | 51.70 | 59.50 | 65.33 | 61.72 | 50.09 | 56.63 | 46.44 | 52.73 | 58.35 | 53.48 | 48.32 | 49.67 | 53.15 |
| Qwen2.5-VL | 52.54 | 59.85 | 65.52 | 64.64 | 51.23 | 57.57 | 48.21 | 56.12 | 60.31 | 56.33 | 52.46 | 52.84 | 55.21 |
| **Online Assistants** | | | | | | | | | | | | | |
| LiveStar | 46.43 | 53.75 | 59.36 | 57.29 | 45.64 | 51.37 | 43.56 | 51.52 | 55.71 | 50.79 | 47.77 | 48.15 | 49.76 |
| LiveStar† | 54.06 | 61.08 | 66.43 | 66.06 | 52.67 | 58.95 | 52.19 | 59.00 | 62.85 | 58.35 | 54.95 | 55.87 | 57.41 |

# D  Online Tasks

Our dataset is organized into 5 distinct online evaluation tasks designed to comprehensively assess Video-LLMs' capabilities in streaming video understanding. Each task targets a specific aspect of temporal reasoning and contextual awareness under causal constraints. Detailed case examples for each task are provided in Tabs. 7, 8, and 9.

---

[2]https://huggingface.co/spaces/yzy666/SVBench

## D.1 Real-time Narration Generation

This task requires Video-LLMs to generate accurate and temporally coherent descriptions for streaming video in real time, where descriptive segments are selectively triggered at transitions between semantic clips to avoid redundancy. Rather than producing isolated captions, the output is expected to form a coherent and continuous narrative that evolves with the visual content. Each segment should be logically connected to the previous ones through transitional phrases or discourse markers (*e.g., first, then, meanwhile, as a result, finally*) to ensure a smooth and natural flow. Coreference resolution is also required: entities introduced earlier (*e.g., "a man"*) may be subsequently referred to using appropriate pronouns (*e.g., "he"*), demonstrating contextual awareness. Although generated in multiple parts, the narration should ultimately come together as a single, fluid, and unified story—one that maintains semantic clarity, narrative cohesion, and temporal alignment with the unfolding video.

## D.2 Online Temporal Grounding

This task requires Video-LLMs to detect and localize temporal segments in a streaming video that semantically correspond to a given textual query, enabling real-time moment retrieval without access to future frames. For each query, the task involves identifying the start point where the visual content begins to align with the described event and the end point where the scene transitions to a different semantic clip. Importantly, multiple distinct segments in a video may independently satisfy the same query—for example, the phrase *"Video of a car in travel."* might correspond to several non-contiguous intervals throughout a long video, each relevant interval should be detected independently and promptly as it unfolds. It assesses a model's capacity for fine-grained, online temporal localization and its ability to detect multiple instances of the same semantic clip over time.

Since current Video-LLMs lack native online temporal grounding capability, we propose a surrogate evaluation strategy. We first divide the dataset based on the length of the relevant window (*e.g., 10s, 20s, 30s*), forming candidate temporal intervals of varying durations. For each video and query pair, we compute the model's per-frame perplexity (*PPL*) with respect to the query, treating lower perplexity as an indicator of stronger semantic alignment. Among the candidate intervals of the specified length, we select the one with the lowest average perplexity as the predicted segment. Finally, we compare the predicted segment with the ground-truth relevant window to compute a temporal deviation metric (*TimDiff*) that quantifies alignment accuracy.

## D.3 Frame-level Dense Question-Answering

This task requires Video-LLMs to generate multiple temporally grounded answers to a single persistent question throughout a video, producing dense yet non-redundant responses that are triggered only by meaningful semantic changes in the visual content. For instance, given the question *"How many people are there in the video now?"*, the model should initially respond with *"One person"* and remain silent until a second individual appears, at which point the response should be updated to *"Two people"*. The task emphasizes the production of sparse but informative answer sequences that reflect event-driven and temporally consistent reasoning. It ensures that an online video model's outputs align closely with significant shifts in the video's semantic clips, prioritizing relevance and conciseness over frequency.

## D.4 Contextual Online Question-Answering

This task requires Video-LLMs to perform contextual question answering on a continuously streaming video by addressing queries issued at arbitrary time points, based on the visual content observed up to that moment. These questions are organized into QA chains—temporally distributed, multi-turn interactions grounded in the same video segment, which may exhibit implicit dependencies between successive questions and answers. Each question must be answered using only the video frames that have been observed at the time of the query, without access to future information. As the video progresses, new questions may arise, which may exhibit contextual dependencies on both historical video segments and dialogues. For example, given the QA chain *Q1: "Who is the competitor in this event?"*, a follow-up question *Q2: "What did he continue to do?"* relies on coreference resolution, where *"he"* refers to the previously identified *"competitor"*. The task emphasizes incremental understanding, memory retention, and context-aware reasoning under causal constraints. It evaluates

the ability to utilize historical information and engage in coherent multi-turn dialogue within a real-time video stream.

### D.5 Multi-turn Interactive Question-Answering

This task requires Video-LLMs to perform multi-turn question answering on a continuously stream-ing video by responding to multiple queries—potentially issued simultaneously at the same time point—based on the visual content observed up to that moment. Similar to the **Contextual Online Question-Answering Task** D.4 , all queries are organized into QA chains, with each chain repre-senting a temporally distributed sequence of questions and answers pertaining to the same semantic segment of the video. In this interactive setting, multiple QA chains may evolve in parallel, and the questions within or across chains may exhibit implicit contextual dependencies, requiring the model to reason jointly over the video history and multiple dialogue threads. The task emphasizes the ability to handle concurrent multi-turn interactions in a streaming context, assessing whether the model can track and resolve complex inter-question relationships while maintaining temporal grounding and causal consistency.

Table 7: Case examples of OmniStar (RNG, OTG, COQ, MIQ).

| Online Task | Case Examples |
|---|---|
| **Real-time Narration Generation** | *<RNG task prompt> Please describe the content of the video in real time and use cohesive words to keep the narrative coherent.*
00:00: In a room, there is a woman with short purple hair wearing a black top, and there are cartoon characters above her head.
00:01 ~ 00:03: Silent.
00:04: Then the cartoon characters above her head keep changing, and finally stay on a cartoon character of a little red man.
00:05 ~ 00:10: Silent.
00:11: Afterwards, the woman plays the cartoon character. She wears a black top hat decorated with a row of white teeth and a red suit against a red background.
00:12 ~ 00:20: Silent.
00:21: Finally, the woman holds a white balloon with a black mouth and eyes drawn on it. She keeps staring at the balloon, and the picture zooms in on her eyes. |
| **Online Temporal Grounding** | *<OTG task prompt> A man reporting from outside of a destroyed house.*
00:00 ~ 01:13: Slient. (Not related)
01:14: Start.
01:15 ~ 01:23: Slient. (Related)
01:24: End.
01:25 ~ 01:29: Slient. (Not related)
01:30: Start.
01:31 ~ 01:39: Slient. (Related)
01:40: End.
01:41 ~ 02:00: Slient. (Not related) |
| **Contextual Online QA** | *<COQ task prompt>*
00:00 ~ 00:16: Silent.
00:17: *Q: What were the two ladies talking about at the beginning of the video clip?*
       A: They were discussing entering the house.
00:18 ~ 00:22: Silent.
00:23: *Q: What are they discussing?*
       A: They are discussing how to express gratitude for each other's help.
00:24 ~ 00:33: Silent.
00:34: *Q: Why are they talking about the topic of gratitude?*
       A: They are talking about the topic of gratitude because one of them helped the other, prompting this discussion. |
| **Multi-turn Interactive QA** | *<MIQ task prompt>*
00:00 ~ 00:06: Silent.
00:07: *Q1: What appears at the beginning of the video?*
       A1: A row of bright windows appears.
       *Q2: What is above the window?*
       A2: It's a white roof.
       *Q3: What is in front of it?*
       A3: There are several white tables.
00:08 ~ 00:14: Silent.
00:15: *Q1: Where is this row of windows located?*
       A1: Located within an indoor space.
       *Q2: What is inside this indoor space?*
       A2: There is a lady and some equipment.
       *Q3: What does the equipment include?*
       A3: Includes yoga mats and fitness balls.
00:16 ~ 00:27: Silent.
00:28: *Q1: What is there at the beginning of the video?*
       A1: There is a room for fitness and exercise.
       *Q2: What are the equipments in the indoor space?*
       A2: There are yoga mats, fitness balls, dumbbells, and other exercise equipment.
       *Q3: What other furniture and decorations are in the room?*
       A3: In the room, there are also mirrors, potted plants, and a basket among other decorations. |

Table 8: Case examples of OmniStar (FDQ-1).

| Online Task | Case Examples |
|---|---|
| **Frame-level Dense QA** | *<FDQ task prompt> What is the person showing to the camera?*
00:00: The person shows nothing to the camera.
00:01 ~ 00:05: Silent.
00:04: The person is showing a white cup to the camera.
00:07 ~ 00:10: Silent.
00:11: The person is showing a grey cup to the camera.
00:12 ~ 00:16: Silent.
00:17: The person is showing a red book to the camera.
00:18 ~ 00:23: Silent. |
| | *<FDQ task prompt> How many times did the person launch the toy car on the slanted plane?*
00:00: The person did not launch the toy car on the slanted plane.
00:01 ~ 00:04: Silent.
00:05: The person launched the toy car once on the slanted plane.
00:06 ~ 00:12: Silent.
00:13: The person launched the toy car twice on the slanted plane.
00:14 ~ 00:19: Silent.
00:20: The person launched the toy car three times on the slanted plane.
00:21 ~ 00:30: Silent. |
| | *<FDQ task prompt> What repeated actions did the person do?*
00:00: The person did not do repeated actions.
00:01 ~ 00:16: Silent.
00:17: The person clapped hands.
00:18 ~ 00:21: Silent.
00:22: The person tapped the table with a ruler.
00:23 ~ 00:27: Silent.
00:28: The person folded paper.
00:29 ~ 00:34: Silent.
00:35: The person did not do repeated actions.
00:36 ~ 00:45: Silent. |
| | *<FDQ task prompt> What letters is the person writing on the paper?*
00:00: Not writing.
00:01 ~ 00:07: Silent.
00:08: The person is writing the letter 'B'.
00:09 ~ 00:14: Silent.
00:15: The person is writing the letter 'E'.
00:16 ~ 00:18: Silent.
00:19: The person is writing the letter 'D'.
00:20 ~ 00:23: Silent.
00:24: Not writing.
00:25 ~ 00:41: Silent. |
| | *<FDQ task prompt> What is the order of the letters on the table?*
00:00: The order of the letters on the table is 'LOVE'.
00:01 ~ 00:18: Silent.
00:19: The letters on the table is being shuffled.
00:20 ~ 00:42: Silent.
00:43: The order of the letters on the table is 'VOLE'.
00:44 ~ 01:01: Silent. |

Table 9: Case examples of OmniStar (FDQ-2).

| Online Task | Case Examples |
|---|---|
| | *<FDQ task prompt> How many times did the objects in the video collide?*
00:00: No collision occurred.
00:01 ~ 00:04: Silent.
00:05: One collision occurred.
00:06 ~ 00:10: Silent.
00:11: Two collisions occurred.
00:12 ~ 00:19: Silent. |
| | *<FDQ task prompt> How many objects are there in the video?*
00:00: There are four objects in the video.
00:01 ~ 00:02: Silent.
00:03: There are five objects in the video.
00:03 ~ 00:07: Silent.
00:08: There are six objects in the video.
00:09 ~ 00:19: Silent. |
| | *<FDQ task prompt> How many moving objects are in the video?*
00:00: No objects are moving in the video.
00:01 ~ 00:03: Silent.
00:04: Two objects are moving in the video.
00:05 ~ 00:08: Silent.
00:09: Three objects are moving in the video.
00:10 ~ 00:19: Silent. |
| **Frame-level Dense QA** | *<FDQ task prompt> What is the scene in the video like?*
00:00: The scene in the video shows several people in the interrogation room of the police station.
00:01 ~ 00:15: Silent.
00:16: The scene in the video changes to five women outside in a garden.
00:17 ~ 00:30: Silent. |
| | *<FDQ task prompt> How many people are wearing glasses in the video now?*
00:00: There is a person wearing glasses in the video.
00:01 ~ 00:08: Silent.
00:09: There are no people wearing glasses in the video.
00:10 ~ 00:15: Silent.
00:16: There are two people wearing glasses in the video.
00:17 ~ 00:21: Silent. |
| | *<FDQ task prompt> What actions is the person performing in the video during the cooking preparation?*
00:00 ~ 00:03: Silent.
00:04: The person takes out a cucumber wrapped in plastic.
00:05 ~ 00:08: Silent.
00:09: The person takes out a knife.
00:10 ~ 00:15: Silent.
00:16: The person takes out a cutting board.
00:17 ~ 00:22: Silent.
00:23: The person takes out a plate. |

# E   Online Evaluation Metrics

We utilize GPT-4o to evaluate the generated answers in terms of semantic correctness and summary fluency.

## E.1   Semantic Correctness

The semantic correctness evaluation is used to assess the ability of Video-LLMs to understand video segments after segmenting the complete video, focusing on whether the model can accurately capture and comprehend all the information in the video segments. When evaluating the semantic correctness of the answers, we input both the model-generated answers for the corresponding video segments

and the ground truth into GPT-4o, and ask GPT-4o to evaluate the model-generated answers from metrics of Semantic Accuracy, Language Quality and Information Completeness, then output scores for these three metrics.

- **Semantic Accuracy.** It evaluates the alignment of core meaning between generated answers and ground truth, evaluating semantic compatibility without errors or hallucinations. The evaluation metric focuses on assessing the overall correctness of the generated answers. The more accurate and closer the generated answer is to the corresponding parts of the ground truth, the higher the score, and vice versa. For this metric, we design five scoring criteria: a score of 10 indicates perfect matching with ground truth and no deviation; a score of 7-9 indicates minor semantic differences but core information intact; a score of 4-6 indicates significant errors or omissions in key details; a score of 1-3 indicates mostly incorrect or irrelevant; a score of 0 indicates completely wrong or unrelated.

- **Language Quality.** It evaluates grammatical correctness, lexical precision, and readability, ensuring clarity and fluency in expression. If the generated answer has less grammatical issues or spelling errors, the score will be higher; otherwise, the score will be lower. For this metric, we design five scoring criteria: a score of 10 indicates flawless grammar, precise wording and no redundancy; a score of 7-9 indicates minor grammatical slips but do not affect understanding; a score of 4-6 indicates obvious grammar issues or unclear phrasing; a score of 1-3 indicates chaotic language and multiple errors hinder comprehension; a score of 0 indicates unintelligible or symbol-filled.

- **Information Completeness.** It evaluates the breadth and depth of information coverage relative to ground truth, including key events and dynamic actions without omissions. This evaluation metric focuses on assessing whether the generated answer is sufficiently comprehensive compared to the ground truth. The more ground truth information the generated answer covers, the higher the score, and vice versa. For this metric, we design five scoring criteria: a score of 10 indicates including all key information from ground truth; a score of 7-9 indicates main points covered but missing minor details; a score of 4-6 indicates partial key information missing but retaining the core content; a score of 1-3 indicates only fragments of information retained; a score of 0 indicates no valid information provided.

## E.2 Summarize Fluency

The summary fluency evaluation is used to assess whether the model, after segmenting the complete video, can notice the information contained in previous video segments and thus incorporate this information into its answers. When evaluating the summary fluency of the answers, we concatenate the model-generated responses for all video segments and the corresponding ground truth for different segments, then input them into GPT-4o, and request GPT-4o to evaluate the model-generated answers from metrics of Writing Logicality, Language Fluency, Writing Conciseness, Semantic Consistency and Narrative Completeness, then output scores for these five metrics.

- **Writing Logicality.** It evaluates the clarity and strength of causal, temporal, or inferential relationships between sentences, ensuring coherent progression of ideas. The clearer and more coherent the cause-and-effect logic in the generated answer, the higher the score it receives. Conversely, if the logic is confused, the score will be lower. For this metric, we design five scoring criteria: a score of 10 indicates flawless logic with no gaps or contradictions; a score of 7-9 indicates minor logical jumps but generally coherent; a score of 4-6 indicates weak connections requiring reader inference; a score of 1-3 indicates frequent illogical breaks or contradictions; a score of 0 indicates no logical structure.

- **Language Fluency.** It evaluates precision in wording and avoidance of redundant content. If the generated answer contains sentences with abrupt transitions, the score will be low. If the transitions are smooth, the score will be high. For this metric, we design five scoring criteria: a score of 10 indicates effortless flow with no awkward phrasing; a score of 7-9 indicates smooth overall with rare hiccups; a score of 4-6 indicates choppy transitions and some re-reading needed; a score of 1-3 indicates relies heavily on forced connectors; a score of 0 indicates unreadable due to disjointedness.

- **Writing Conciseness.** It evaluates precision in wording and avoidance of redundant content. If the generated answer contains redundant or repetitive expressions, the score will be low.

If the writing is concise and succinct, the score will be high. For this metric, we design five scoring criteria: a score of 10 indicates precise wording and no redundancy; a score of 7-9 indicates minor repetitions but efficient overall; a score of 4-6 indicates verbose sections needing trimming; a score of 1-3 indicates excessive repetition or filler; a score of 0 indicates entirely redundant or irrelevant.

- **Semantic Consistency.** It evaluates uniformity in terminology, pronouns, and tense usage across the text. If the generated answer contains inconsistencies in terminology or pronoun usage, the score will be low. If the pronouns, tenses, and terminology are consistent throughout the answer, the score will be high. For this metric, we design five scoring criteria: a score of 10 indicates perfect consistency in all elements; a score of 7-9 indicates slight inconsistencies with no impact; a score of 4-6 indicates multiple inconsistencies requiring correction; a score of 1-3 indicates confusing shifts in terms or perspective; a score of 0 indicates incoherent due to inconsistency.

- **Narrative Completeness.** It evaluates coverage of all key information without omissions. This evaluation metric focuses on assessing whether there is any missing information in the overall narrative of the generated answer. The more information is missing, the lower the score; conversely, the less information is missing, the higher the score. For this metric, we design five scoring criteria: a score of 10 indicates including all critical details; a score of 7-9 indicates covering main points but having minor gaps possible; a score of 4-6 indicates missing some key elements; a score of 1-3 indicates only fragmented or irrelevant information; a score of 0 indicates no relevant information.

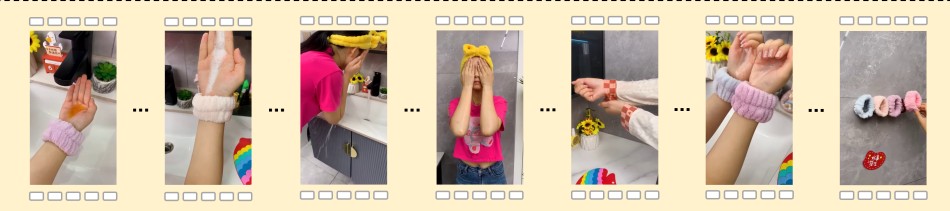

**Timestamp: 00:00**

**Ground Truth:** In a bathroom, a pair of hands are placed under a black faucet, with orange liquid on them and a light purple fluffy scrunchie on the wrist. On the washstand on the left side of the picture, there is a red ornament with a white duckling and a sign on the ornament that says "Keep going today". On the washstand on the right side of the picture, there is a small potted plant with greenery.

**VideoLLM-Online:** You hold a bottle.

**MMDuet:** Slient. (nothing output)

**LiveStar:** A hand wearing a pink wristband is placed on a white table. The wristband has the words "Sunny Day" written on it in black. There are some items on the table, including a small white duck-shaped object and a pot of green plants.

**Timestamp: 00:02**

**Ground Truth:** Then the hand sweeps upward across a black faucet, and the shot changes. White foam flows down from the upright hand in the picture, flowing down to the light white plush wristband on the wrist.

**VideoLLM-Online:** You rub your hands.

**MMDuet:** Slient. (nothing output)

**LiveStar:** Then, the hand wearing the pink wristband is placed under a running faucet. The water from the faucet washes over the white liquid on the palm, gradually washing it away.

**Timestamp: 00:04**

**Ground Truth:** Immediately afterward, the camera pans to a woman wearing a red shirt and a yellow headband, squatting in front of the sink, washing her face.

**VideoLLM-Online:** You rub your hands.

**MMDuet:** Apply the cream to the face.

**LiveStar:** Then the scene then switches to a bathroom. A woman wearing a pink T-shirt is standing in front of the sink, washing her face. The water from the faucet flows down the drain.

**Timestamp: 00:06**

**Ground Truth:** Then the camera switches again. A left hand reaches out to the other side of the screen, and the other right hand is arranging the cuff of the left hand. The clothes are beige, and the cuffs have a pink and white checkered pattern.

**VideoLLM-Online:** You rub your hands.

**MMDuet:** Slient. (nothing output)

**LiveStar:** The scene then switches to the woman's hands. She is wearing pink and white wristbands, and one hand is adjusting the cuff of the other.

**Timestamp: 00:08**

**Ground Truth:** Finally, the camera turns, and four different colors of plush wristbands are shown in the picture. The four plush wristbands are neatly arranged on the bathroom wall.

**VideoLLM-Online:** You rub your hands.

**MMDuet:** Slient. (nothing output)

**LiveStar:** The scene then switches to a gray wall, where the woman is arranging four wristbands of different colors.

**Comment:** This case shows that VideoLLM-Online and MMDuet, perform poorly on the RNG task. VideoLLM-Online's answers are brief, limited to a first-person perspective (*"You ..."*), and often redundant, while MMDuet misses key points and provides simple, inaccurate responses. Compared to these, LiveStar exhibits a more timely and precise understanding of the video and sometimes captures finer details beyond the annotations (e.g., timestamp 00:04).

Figure 5: **Comparison of VideoLLM-online, MMDuet, and LiveStar on the RNG task.**

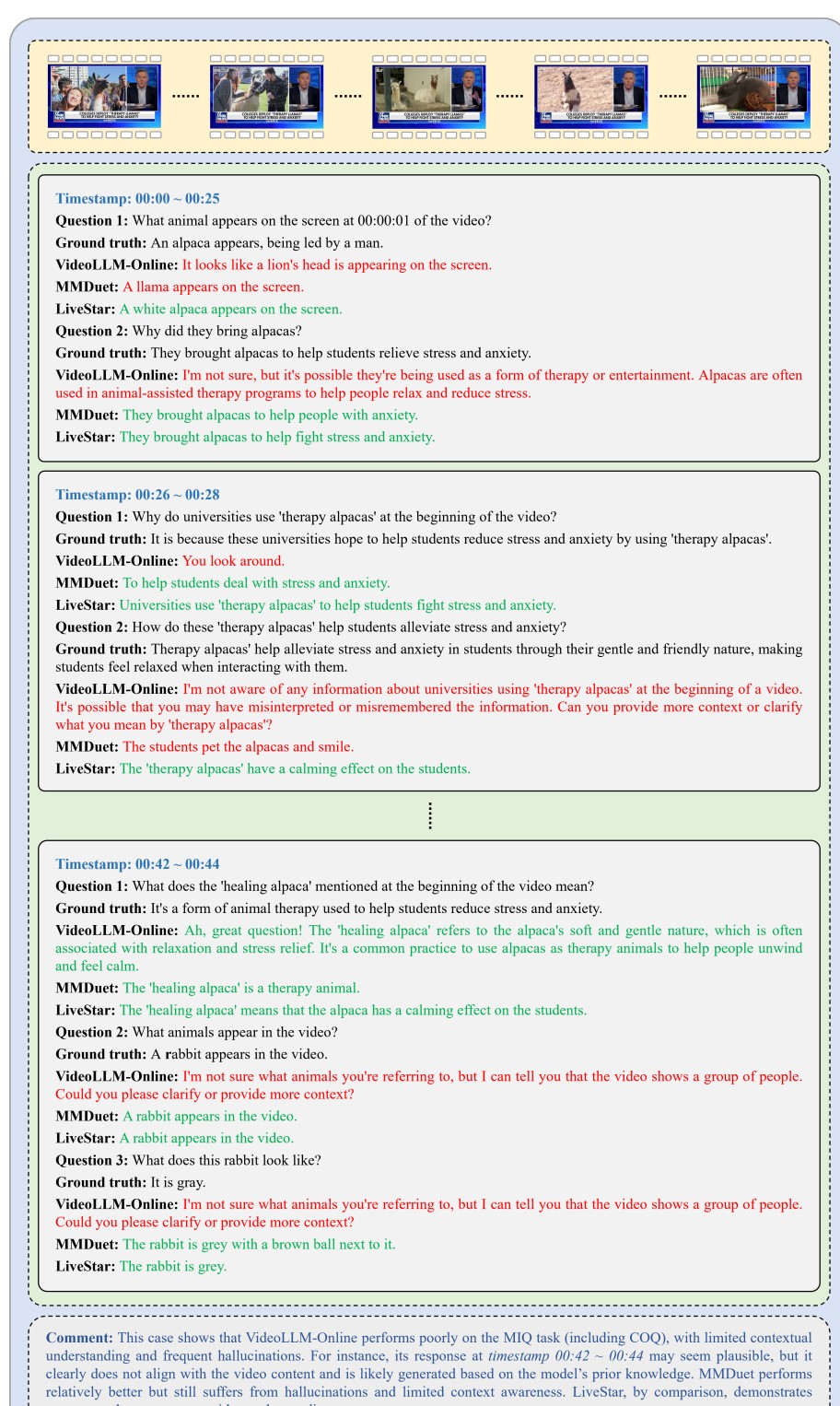

Figure 6: **Comparison of VideoLLM-online, MMDuet, and LiveStar on the MIQ task (also representative of the COQ task).**

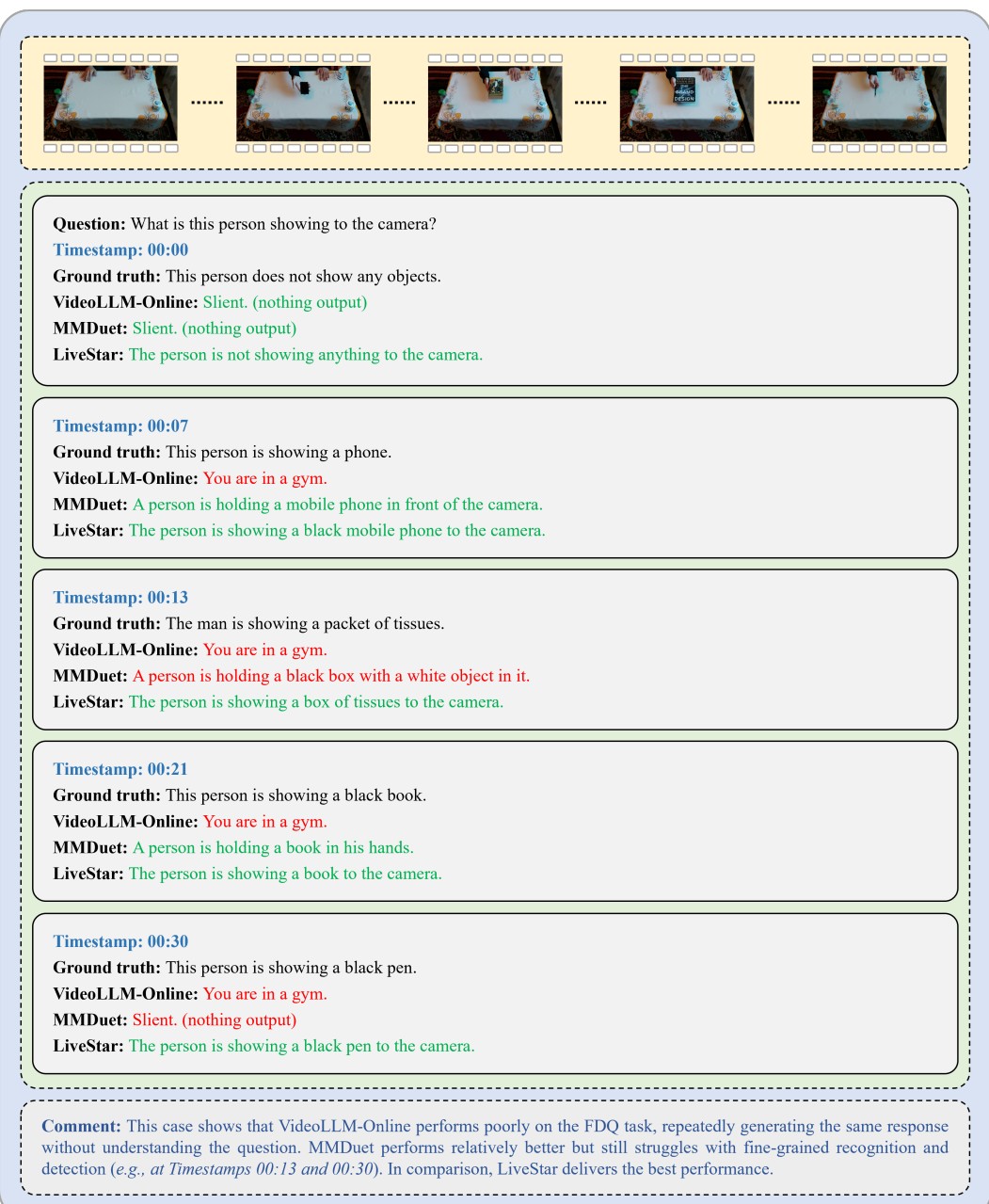

Figure 7: **Comparison of VideoLLM-online, MMDuet, and LiveStar on the FDQ task.**

# F    Visualization of LiveStar

We develop a web-based demo for LiveStar using Gradio, which enables users to upload arbitrary videos and perform real-time streaming chat. The interface also allows flexible adjustment of inference parameters (e.g., the tunable scaling factor $\alpha$) to accommodate various application scenarios. The demo interface is illustrated in Figure 8.

# G    Impact Statements

Our proposed LiveStar, a pioneering live streaming assistant with temporal decision-making capabilities, presents significant societal implications. On the positive side, LiveStar introduces a paradigm

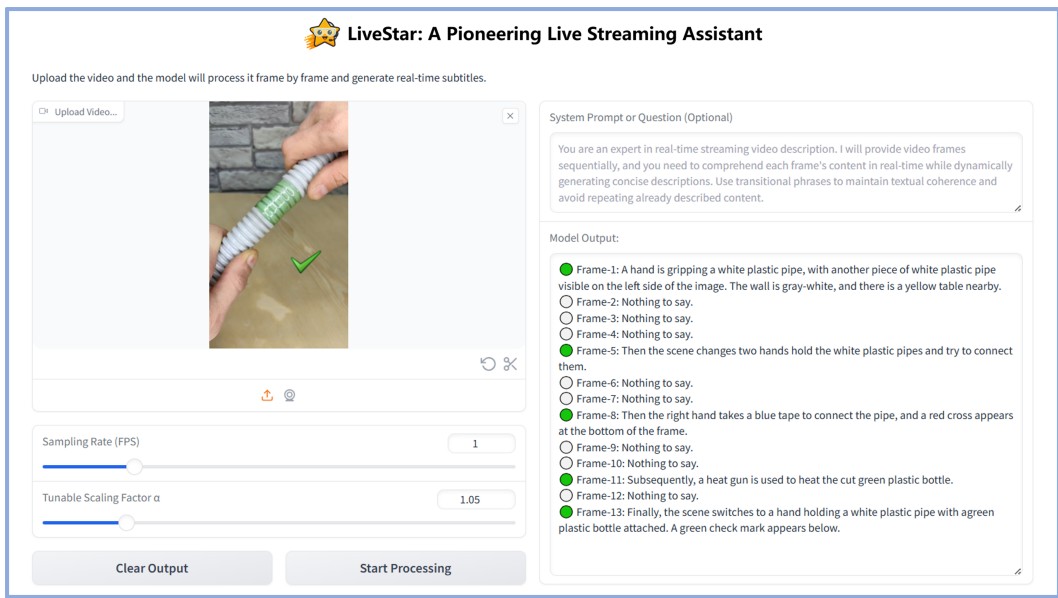

Figure 8: **A web demo of LiveStar.** Green points indicate generated outputs, while white denotes no output.

shift in online understanding by dynamically determining when and how to generate outputs. This enables efficient processing of real-time online data with reduced latency and computational overhead. This innovation holds promise for applications such as real-time online platforms, where it could enhance viewer engagement through instant content summarization, automated moderation, or adaptive subtitling. Additionally, its ability to prioritize critical events in surveillance systems could improve public safety by enabling faster emergency responses in scenarios like traffic monitoring or crowd management. The technology also has potential in assistive tools for individuals with disabilities, such as real-time visual interpretation or context-aware navigation aids.

However, the technology's adaptive decision logic also introduces risks. When integrated into drones or public cameras, it raises concerns about mass privacy infringement, particularly if deployed without transparent governance. In conclusion, while LiveStar advances real-time online video understanding, its societal adoption demands robust governance frameworks. We advocate for strict regulatory compliance in privacy-sensitive domains and interdisciplinary collaboration to mitigate misuse risks while maximizing its benefits for human-centric applications.

## H   Prompt

**Prompt to Evaluate Semantic Correctness Using GPT-4o Score**

Task Description:
You are an expert in evaluating captions generated for various scenes in videos. I will provide you with the ground truth for each scene caption and the caption generated by the model. You need to score the given caption generated by the model based on following three dimensions and output the respective scores for each dimension:
1. Semantic Accuracy: Measures the match in core meaning between the generated caption and Ground Truth.
Scoring criteria: - 10: Perfect matching with Ground Truth, no deviation. - 7-9: Minor semantic differences but core information intact. - 4-6: Significant errors or omissions in key details. - 1-3: Mostly incorrect or irrelevant. - 0: Completely wrong or unrelated.
2. Language Quality: Evaluates grammatical correctness, word choice, and readability.
Scoring criteria: - 10: Flawless grammar, precise wording and no redundancy. - 7-9: Minor grammatical slips, but do not affect understanding. - 4-6: Obvious grammar issues or unclear phrasing. - 1-3: Chaotic language, and multiple errors hinder comprehension. - 0: Unintelligible or symbol-filled.
3. Information Completeness: Assesses breadth and depth of information coverage relative to Ground Truth.
Scoring criteria: - 10: Including all key information from Ground Truth. - 7-9: Main points covered, but missing minor details. - 4-6: Partial key information missing, but retaining the core content. - 1-3: Only fragments of information retained. - 0: No valid information provided.
Requirements:
1. I will give you the ground truth with "Ground Truth:" and the model-generated caption with "Model Response:".
2. When outputting the score for the Semantic Accuracy dimension, it must start with "Semantic Accuracy:". When outputting the score for the Language Quality dimension, it must start with "Language Quality:". When outputting the score for the Information Completeness dimension, it must start with "Information Completeness:". Apart from the scores for the three dimensions, no other characters are allowed in the output.
Ground Truth: {ground_truth}
Model Response: {model_response}

## Prompt to Evaluate Semantic Correctness Using GPT-4o Score

Task Description:
You are an expert in evaluating video content caption generation. I will provide you with the ground truth of video caption and the caption generated by the model. You need to score the given caption generated by the model based on following five dimensions and output the respective scores for each dimension:
1. Writing Logicality: Evaluates causal, temporal, or inferential relationships between sentences.
Scoring criteria: 10: Flawless logic with no gaps or contradictions. 7-9: Minor logical jumps but generally coherent. 4-6: Weak connections requiring reader inference. 1-3: Frequent illogical breaks or contradictions. 0: No logical structure.
2. Language Fluency: Assesses natural transitions between sentences.
Scoring criteria: 10: Effortless flow with no awkward phrasing. 7-9: Smooth overall with rare hiccups. 4-6: Choppy transitions, some re-reading needed. 1-3: Relies heavily on forced connectors. 0: Unreadable due to disjointedness.
3. Writing Conciseness: Measures brevity and avoidance of redundancy.
Scoring criteria: 10: No redundancy; precise wording. 7-9: Minor repetitions but efficient overall. 4-6: Verbose sections needing trimming. 1-3: Excessive repetition or filler. 0: Entirely redundant or irrelevant.
4. Semantic Consistency: Checks uniformity in terminology, pronouns, and tense.
Scoring criteria: 10: Perfect consistency in all elements. 7-9: Slight inconsistencies with no impact. 4-6: Multiple inconsistencies requiring correction. 1-3: Confusing shifts in terms or perspective. 0: Incoherent due to inconsistency.
5. Narrative Completeness: Evaluates whether the caption covers all key information.
Scoring criteria: 10: Including all critical details. 7-9: Covering main points, but having minor gaps possible. 4-6: Missing some key elements. 1-3: Fragmented or irrelevant info only. 0: No relevant information.
Requirements:
1. I will give you the ground truth with "Ground Truth:" and the model-generated caption with "Model Response:".
2. When outputting the score for the Writing Logicality dimension, it must start with "Writing Logicality:". When outputting the score for the Language Fluency dimension, it must start with "Language Fluency:". When outputting the score for the Writing Conciseness dimension, it must start with "Writing Conciseness:". When outputting the score for the Semantic Consistency dimension, it must start with "Semantic Consistency:". When outputting the score for the Narrative Completeness dimension, it must start with " Narrative Completeness:". Apart from the scores for the five dimensions, no other characters are allowed in the output.
Ground Truth: {ground_truth}
Model Response: {model_response}

**Prompt to Generate Second-level Caption**

You are an expert in generating descriptions of streaming videos. Now, given all the frames of a 1-second video, analyze the visual content and generate complete subtitles.

Input: real-time frames of a video clip or video stream.

Steps: 1. Analyze the video content: identify the main events, characters, scenes, and actions in the video. 2. Extract key information: extract key information from the video that can represent the video content. 3. Generate description: generate a concise and rich text description based on the extracted information. 4. Proofreading and optimization: check the accuracy and fluency of the description to ensure that it conforms to the video content.

Notes: 1. Make sure the description is directly related to the video content. 2. The description is prohibited from using modifiers such as adjectives and adverbs. 3. Avoid verbosity and keep it concise. 4. Pay attention to the logic and coherence of the description. 5. It is forbidden to mention content that is not mentioned in the video.

Output example: "A basketball player (wearing a red jersey) jumps vertically in the middle of the court under the bright court lights, with his arms reaching towards the net. The ball spins clockwise in the air, while two defensive players (in blue uniforms) prepare to grab the rebound."

---

**Prompt to Generate Real-Time Narration**

Task: Generate fluid, cinematically-coherent real-time narration that creates continuous visual storytelling. Integrate persistent historical elements with instantaneous observations using temporal reasoning and causal chaining.

Inputs: 1. Historical Context (H): {History}
2. Current Second Data (C): {Current_captions}
3. Current Video Frames

Processing Requirements:

1. Dynamic Temporal Modeling: - Maintain object/interaction state vectors across sliding windows - Calculate kinematic signatures (jerk vectors, path integrals) - Anticipate action sequences using momentum calculus - Differentiate transient vs sustained states ("progressively veering" vs "abrupt swerve")

2. Scene State Engine: - Preserve spatial memory lattice for persistent entities - Refresh object kinematics (pose derivatives, interaction potential) - Implement occlusion resilience through Markovian tracking - Maintain projective geometry constraints for spatial coherence

3. Narrative Flow Synthesis: - Forge event domino sequences with causal integrity - Employ aspectual verb framing (sustaining/commencing/culminating) - Apply filmic continuity principles (eyeline matches, temporal ellipses) - Optimize attention allocation through depth-ordered significance

Output Format: "[XX:XX-XX:XX] [Subject][action] [preposition][environment] while [parallel process], with [situational awareness] (phase indicator)"

Example: Input H: ["[00:00-00:05] Red sedan accelerates past stationary truck, establishing eastward movement"] Input C: ["[00:05-00:06] Vehicle decelerating near crosswalk. [00:06-00:07] Pedestrian initiating crossing sequence"]

Output: "[00:05-00:07] The crimson sedan progressively moderates speed approaching a crosswalk, its path converging with a pedestrian now committing to traverse, while the historically logged truck persists in the rear-left zone"

## Prompt to Generate Real-Time Narration

Task: As a real-time narration generator, create vivid and coherent video commentary by combining:

Historical context from previous narration: {Insert past narration}

Current video clip

Per-second captions: {List timestamped captions}

Guidelines:

- Temporal continuity: Maintain strict time synchronization between captions (seconds 00:00-00:XX) and narration
- Context weaving: Blend historical context with new developments while avoiding repetition
- Fluency: Use natural transitions between sentences and temporal markers (e.g., "Meanwhile," "Suddenly")
- Focus: Highlight key actions/objects from captions while maintaining narrative flow

Output Requirements:

- 1-2 sentences per major event/transition
- Active voice and descriptive verbs
- Neutral observatory tone without speculation

Example Output Format: "[00:05-00:07] The crimson sedan progressively moderates speed approaching a crosswalk, its path converging with a pedestrian now committing to traverse, while the historically logged truck persists in the rear-left zone"

