# OpenReview forum: "LiveStar: Live Streaming Assistant for Real-World Online Video Understanding"
_NeurIPS.cc/2025/Conference — NeurIPS 2025 poster_

### Official Review · Reviewer_dLwS · 2025-06-05

**Clarity:** 3
**Significance:** 3
**Originality:** 4
**Rating:** 4
**Confidence:** 3

**Summary:**

This paper introduces LiveStar, a live streaming assistant designed for real-world online video understanding. It addresses the limitations of existing online Video Large Language Models (Video-LLMs) that struggle with simultaneously processing continuous frame-by-frame inputs and determining optimal response timing, which often compromises real-time responsiveness and narrative coherence. LiveStar proposes SCAM to preserve consistency across dynamically evolving frame sequences. And the paper introduces OmniStar, a comprehensive dataset for training and benchmarking online video understanding. Extensive experiments across three benchmarks demonstrate LiveStar's state-of-the-art performance compared to existing online Video-LLMs.

**Questions:**

Please see the weaknesses. If the authors adequately address these concerns in their rebuttal, I will reconsider my score.

**Ethical Concerns:**

["NO or VERY MINOR ethics concerns only"]

**Final Justification:**

After reviewing the performance of Livestar on additional benchmarks and considering the feedback from other reviewers, I maintain my positive score. Further analysis of the results on additional benchmarks requires additional refinement in future versions of this work.

**Limitations:**

yes

**Paper Formatting Concerns:**

There is no formatting issues.

**Quality:**

4

**Strengths And Weaknesses:**

**Strengths:**

1. This paper makes substantial contributions by establishing a comprehensive framework for an online video assistant, encompassing both methodological innovations and dataset construction. And the paper conducts extensive experimental validations that rigorously demonstrate the effectiveness of LiveStar.

2. Unlike previous online Video-LLMs rely on learning the EOS token, this paper proposes SCAM to solve the challenge of response-silence imbalance and consecutive frame inconsistency, which is interesting.

3. LiveStar consistently outperforms other online Video-LLMs, showing significant improvements in semantic correctness and reduced timing differences.

**Weaknesses:**

1. In the ablation of Impact of Response-Silence Threshold, the divergent trends in the three metrics with varying decoding thresholds need a more rigorous theoretical analysis.

2. The calculation methods for the evaluation metrics have not been provided. I want to know the specific computational procedures for these metrics.

3. In your paper, you suggest that the EOS token may lead to pre-training misalignment and vocabulary confusion issues, which appears to be an intuitive conjecture. However, modern large language models (LLMs) possess substantial comprehension capabilities, meaning the EOS token might not necessarily cause these problems or may only marginally impair their conversational performance. Could this issue be empirically investigated through experimentation?

4. In Table 1 (online split), the other models were not trained on the dataset’s training set. Could this introduce potential fairness issues in the comparative evaluation?

---

> ### Author Rebuttal · Authors · 2025-07-31
>
> Many thanks to your valuable comments and questions, which help us a lot to improve our work. We address your questions as follows.
>
> > [W2]The calculation methods for the evaluation metrics have not been provided. I want to know the specific computational procedures for these metrics.
>
> First, We would like to address Weaknesses 2 by introducing the evaluation metric we proposed, as this will help clarify the context and facilitate a better understanding of Weaknesses 1.
>
> For each video, we have:
>
> - A set of ground-truth (GT) semantic clips  $G = \lbrace g_1, g_2, \dots, g_N \rbrace$, where each clip $g_i$ is defined by: a start time $t_{\text{start},i}$, an end time $t_{\text{end},i}$ and a GT caption.
>
> - A set of model-generated responses $R = \lbrace r_1, r_2, \dots, r_M \rbrace$, where each response $r_j$ has an associated response time $t_{\text{resp},j}$.
>
> For each $ g_i \in G $, define the set of matching responses as:
> $$
> M_i = \lbrace r_j \in R \mid t_{\text{resp},j} \in [t_{\text{start},i},\ t_{\text{end},i}] \rbrace
> $$
>
> - If $ |M_i| = 0 $, no response is generated for $ g_i $; in this case, we assign an **empty string** $ r_j = $ ' '  as the placeholder response for evaluation.
> - If $ |M_i| \geq 1 $, we select the **first matching response** $ r_j \in M_i $ (i.e., with minimal $ t_{\text{resp},j} $) as the representative output.
>
> As detailed in Appendix G, SemCor and SumFluen is calculated using GPT-4o as an evaluator. The process is as follows:
>
> 1. **SemCor (Semantic Correctness):** This metric evaluates the quality of an individual response generated for **a specific semantic clip**. We prompt GPT-4o to score the pair $(r_j, g_i)$ on three dimensions (each on a 0–10 scale): *Semantic Accuracy*, *Language Quality*, and *Information Completeness*.
>
>
> 2. **SumFluen (Summarize Fluency):** This metric evaluates the holistic narrative quality of the model’s complete output sequence, treating all responses as a unified story. Let the **full model narrative** be the concatenation of all responses in temporal order:
>     $$
>     \mathcal{N}\_{\text{model}} = \text{Concat}\_{j=1}^M (r_j)
>     $$
>     Similarly, construct the **ground-truth narrative** by concatenating all GT captions in temporal order:
>     $$
>     \mathcal{N}\_{\text{GT}} = \text{Concat}\_{i=1}^N (g_i)
>     $$
>     We evaluate $ \mathcal{N}\_{\text{model}} $ against $ \mathcal{N}\_{\text{GT}} $ by prompting GPT-4o to assess them on five storytelling-relevant dimensions(each on a 0–10 scale): *Writing Logicality*, *Language Fluency*, *Writing Conciseness*, *Semantic Consistency*, and *Narrative Completeness*.
>
> The metrics TimRedun, TimCover and TimDiff are calculated as:
>
> 3. **TimRedun (Timing Redundancy)**
>    Measures the deviation from the ideal of exactly one response per semantic clip:
>
> $$
> \text{TimRedun} = \frac{1}{N} \sum_{i=1}^N \left||M_i| - 1\right|
> $$
>
> 4. **TimCover (Timing Coverage)**
>    Indicates the proportion of semantic clips that were covered by at least one response:
>
> $$
> \text{TimCover} = \frac{1}{N} \sum_{i=1}^N \mathbb{1}[|M_i| > 0]
> $$
>
> 5.  **TimDiff (Timing Difference)**
>     Captures the cumulative response latency for all matched outputs within a semantic clip, while penalizing missed clips by their full duration:
>
> $$
> \text{TimDiff} = \frac{1}{N} \sum_{i=1}^N \left( \mathbb{I}\big[|M_i| = 0\big] \cdot \big(t_{\text{end},i} - t_{\text{start},i}\big) + \mathbb{I}\big[|M_i| > 0\big] \cdot \sum_{r_j \in M_i} \big(t_{\text{resp},j} - t_{\text{start},i}\big) \right)
> $$
>
> > [W1]In the ablation of Impact of Response-Silence Threshold, the divergent trends in the three metrics with varying decoding thresholds need a more rigorous theoretical analysis.
>
> From a theoretical standpoint, an ideal model should **trigger a response only once at the first frame of each semantic clip** and remain silent otherwise. However, in our proposed LiveStar model, the core of our SVeD mechanism lies in its decoding condition:
>
> $$
> \text{PPL}\_{\text{current}} > \alpha \cdot \text{PPL}\_{\text{ref}}
> $$
>
> Here, the tunable scaling factor α directly influences the model's sensitivity to semantic changes. A higher α value raises the perplexity threshold required to trigger a new response, which means the model will wait for a more significant or prolonged semantic shift before acting.
>
> Specifically, when α is small, the model tends to decode more frequently, potentially generating an output at every frame in the extreme case. This leads to an increased number of unnecessary responses within each semantic clip, with responses occurring earlier in time. As a result, the temporal evaluation metrics reflect a **smaller TimDiff** , a **larger TimRedun**, and a **higher TimCover** . In contrast, as α increases, the model becomes more conservative in decoding, resulting in fewer outputs and delayed response timings. In the extreme case, no output may be generated at all. Under such conditions, both **TimRedun and TimCover decrease**, while **TimDiff increases** significantly—particularly when a semantic clip receives no response, leading to a substantial penalty.
>
> The experimental results, computed according to the formulas defined in **Answer 1**, align perfectly with the above theoretical analysis, thereby validating the role of α in regulating the model's response behavior.
>
> > [W3]In your paper, you suggest that the EOS token may lead to pre-training misalignment and vocabulary confusion issues ... Could this issue be empirically investigated through experimentation?
>
> We agree that modern LLMs are remarkably robust and can adapt to various fine-tuning objectives. Our intention was not to suggest that the EOS-based paradigm is fundamentally flawed, but rather to explore an alternative that might better align with the core strengths of vision-language models in a streaming context.
>
> For pre-training misalignment, our core idea is to better align the fine-tuning objective with the model's pre-training. In online video, a majority of frames are silent. The EOS paradigm trains the model to map these rich visual inputs to a single, non-descriptive EOS token. While effective, this differs from the pre-training goal of generating descriptive text. Our SCAM training strategy, by contrast, consistently focuses the model on the descriptive task by only computing loss on meaningful frame-caption pairs. We believe this contributes to LiveStar's strong performance on third-party benchmarks like Ego4D (Table 3), where it surpasses other online assistants.
>
> For vocabulary confusion, we agree that this point requires nuance. Our concern was more practical than theoretical, noting that VideoLLM-online’s implementation used common token `','` (comma) for the EOS role. To avoid overgeneralization, we will remove this claim in our revision and clarify that the issue is implementation-dependent.
>
> Finally, we wish to reiterate our deep respect for the pioneering EOS-based works that inspired this research. Our goal is to propose a complementary approach that contributes to the diversity of methods in this exciting field. We believe that exploring different paradigms is valuable, and we sincerely thank you for helping us improve our work.
>
> > [W4]In Table 1 (online split), the other models were not trained on the dataset’s training set. Could this introduce potential fairness issues in the comparative evaluation?
>
> We strictly separated the training and testing datasets to ensure a fair evaluation protocol. The model was first trained using our proposed SCAM to learn the alignment between visual and linguistic representations, without any exposure to test data during training.
>
> To provide a more direct and fair comparison of the models' intrinsic capabilities, we conducted evaluations on a pre-existing, third-party benchmark where all models are on a level playing field. As shown in Table 3 (Ego4D Narration Stream benchmark), LiveStar significantly outperforms all other online assistants.
>
> Furthermore, we evaluated this version of LiveStar (without fine-tuning on OmniStar) on the OmniStar-RNG task. The results are as follows:
> | **Method** | **Offline Evaluation (Fixed decoding)**  |   |   |   | **Online Evaluation (Response-silence decoding)**  |     |     |     |     |
> |---|---|---|---|---|---|---|---|---|---|
> |  | **PPL↓** | **TokAcc↑** | **SemCor↑** | **SumFluen↑** | **TimDiff↓** | **TimRedun↓** | **TimCover↑** | **SemCor↑** | **SumFluen↑** |
> | **Online Assistants**   |
> | VideoLLM-online | 9.73 | 0.49 | 3.01 | 0.69 | 2.67  | 2.15  | 0.80  | 1.68 | 0.59  |
> | VideoLLM-MoD | 9.93 | 0.48 | 2.89 | 0.65 | 2.54  | 2.49  | **0.90**   | 1.66 | 0.55  |
> | MMDuet | 5.69 | 0.60 | 4.29 | 3.40 | 2.32  | **0.62**   | 0.51  | 1.93 | 2.69  |
> | **LiveStar (Ego4D verison)** | 5.48 | 0.60 | 4.35 | 4.20 | 1.93 | 1.12  | 0.67  | 2.85 | 3.91 |
> | **LiveStar (Ours)** | **5.14** | **0.62**  | **4.62** | **4.55** | **1.91** | 0.95 | 0.71 | **3.19** | **4.25** |
>
> If you are interested in LiveStar's performance on additional online (e.g., OVBench) or offline (e.g., MVBench, VideoMME, and LVBench) benchmarks, please refer to our response to Reviewer wXqt and j4Ki, where we provide further evaluation results and analysis across a broader set of tasks.
>
> ---
> Thank you once again for your thoughtful and constructive reviews. We have carefully considered all your comments and will incorporate these changes into the revised version. We hope that our responses and planned revisions have addressed your concerns, and we would be very appreciative if you might reconsider your score of our work.

---

> > ### Comment · Reviewer_dLwS · 2025-08-05
> >
> > Thanks for your response! It has addressed most of my concerns, and the introduction of evaluation metrics are comprehensive. After reviewing the performance of Livestar on additional benchmarks and considering the feedback from other reviewers, I maintain my positive score. Further analysis of the results on additional benchmarks requires additional refinement in future versions of this work.

---

> > > ### Author Response · Authors · 2025-08-08
> > > **Grateful for Your Positive Support of LiveStar!**
> > >
> > > Thank you for your insightful review! We confirm that further benchmark analysis will be incorporated into the camera-ready version. We are thrilled that you highlighted major advances: our comprehensive framework for online video assistants, covering both innovative methods (SCAM) and dataset construction. You found SCAM a smart fix for key video challenges, moving beyond old EOS tricks. Best of all, you highlighted how LiveStar clearly outperformed other models in tests, boosting accuracy and timing. You confirmed our response fixed most issues, loved the thorough evaluation metrics, and are sticking with your acceptance recommendation. With utmost respect for your expertise!

---

### Official Review · Reviewer_aC31 · 2025-06-24

**Clarity:** 3
**Significance:** 3
**Originality:** 3
**Rating:** 5
**Confidence:** 3

**Summary:**

The paper introduces LiveStar, an online video understanding framework that addresses key limitations of existing Video-LLMs, such as poor real-time response timing and narrative inconsistency. LiveStar employs adaptive streaming decoding with: (1) a training strategy using Streaming Causal Attention Masks for incremental video-language alignment, (2) a Streaming Verification Decoding framework to determine optimal response timing, and (3) memory-aware acceleration via peak-end compression and KV caching. The authors also present OmniStar, a comprehensive dataset covering real-world scenarios. Experiments show LiveStar achieves state-of-the-art performance compared to other methods. Key contributions include a novel response-silence paradigm, a diverse benchmark, and practical deployment capabilities for long-video streams.

**Questions:**

1. The SVeD decoding threshold parameter α is shown to be sensitive. Is there a way to dynamically learn or adapt this threshold rather than fixing it empirically?

2. The perplexity is used to determine whether a response should be generated, then a complete response must be produced at each forward pass. Wouldn't this significantly slow down the inference speed, since generating a full response involves producing multiple tokens, whereas generating a single [EOS] token allows for much faster termination?

**Ethical Concerns:**

["NO or VERY MINOR ethics concerns only"]

**Final Justification:**

In my opinion, this paper does not have any obvious flaws that would lead to a direct rejection. After carefully reading the author's response, I believe that although LiveStar still has some flaws, the relatively weak evaluation in the initial version has become quite solid after the reviewers' Rebuttal, and the active response design in this paper provides a new idea for such models. Althougth the performance improvement it achieves and the novelty of its approach are not particularly high, it also exceeds the acceptance bar.

**Limitations:**

Yes, in appendix

**Quality:**

3

**Strengths And Weaknesses:**

## Strengths
1. The technical contributions are well-motivated and clearly articulated. LiveStar effectively integrates innovations in training and inference to enable efficient, coherent, and temporally appropriate responses in streaming video settings.

2. Using perplexity instead of having the model explicitly generate an EOS token to decide whether to update the response is a brilliant idea. It effectively transforms the proactive response problem, thereby mitigating the disruption to the original capabilities of the MLLM.

## Weakness
1. The authors have only demonstrated the effectiveness of their approach on their proposed benchmark and SVBench. It is recommended to provide evaluation metrics on additional online video benchmarks (e.g., OVBench) and offline video benchmarks (e.g., videomme) to better illustrate the generalizability of the method.

2. Compressing frames into 16 tokens may miss subtle visual cues (e.g., small object interactions, fine-grained actions), as acknowledged in the limitations.

3.  A direct comparison with recent models like StreamMind, LION-FS and VideoChat-Online would better position LiveStar.

---

> ### Author Rebuttal · Authors · 2025-07-31
>
> Many thanks to your valuable comments and questions, which help us a lot to improve our work. We address your questions as follows.
>
> > [W1] The authors have only demonstrated the effectiveness of their approach on their proposed benchmark and SVBench... It is recommended to provide ... additional online video benchmarks (e.g., OVBench) and offline video benchmarks (e.g., videomme) ...
>
> We appreciate the reviewer's suggestion for broader benchmarking. Our initial evaluation already spanned **three distinct benchmarks** designed to cover a variety of online video understanding scenarios: our proposed OmniStar benchmark (Tables 1 & 2), Ego4D Narration (Table 3), and SVBench (Table 6). These benchmarks collectively assess the core capabilities of proactive, real-time response generation that are central to our work.
>
> Nevertheless, we fully agree that validation on additional standard benchmarks is important. Following your suggestion, we have conducted new experiments on both online and offline benchmarks to further demonstrate the generalizability of LiveStar.
>
> **Performance on Additional Online Benchmark (OVBench):**
> we conducted additional experiments on OVBench: LiveStar significantly outperforms VideoLLM-online, MovieChat, and Flash-VStream (reported in the original paper), though it trails VideoChat-Online marginally. We attribute this to two factors:
>
> - LiveStar optimizes for *text-generation tasks* (e.g., narration, QA), while OVBench emphasizes multiple-choice QA.
> - OVBench evaluates *passive-response models* (no output-timing decisions), whereas LiveStar and VideoLLM-online are *active-response* systems.
>   Task objectives thus differ, and we will discuss this nuance in the revision.
>
> | Model   | FP  |  |  | THV |  |  | PM  |  |  | SP  |  | STP |  | TP  |  |  | Avg |
> | --- | --- | --- | --- | --- | --- | --- | --- | --- | --- | --- | --- | --- | --- | --- | --- | --- | --- |
> |   | AA  | GSP | MP  | AP  | SV  | OP  | AR  | PR  | TR  | AL  | OP  | AT  | OT  | AS  | SL  | OES |  |
> | VideoLLM-online  | 0.00   | 1.80   | 20.90  | 5.20   | 5.90   | 32.60  | 0.00   | 2.30   | 26.70  | 0.60   | 26.60  | 0.90   | 19.90  | 0.90   | 1.70   | 8.30   | 9.60   |
> | MovieChat  | 23.10  | 27.50  | 23.60  | 58.40  | 43.90  | 40.30  | 25.60  | 31.10  | 23.90  | 26.90  | 39.60  | 24.40  | 28.90  | 29.30  | 25.50  | 21.90  | 30.90  |
> | Flash-VStream | 26.90  | 37.60  | 23.90  | 60.10  | 41.90  | 40.00  | 23.40  | 35.30  | 26.10  | 24.70  | 28.80  | 27.00  | 21.40  | 29.80  | 25.60  | 26.80  | 31.20  |
> | LiveStar   | 37.50  | 57.06  | **25.00** | 48.21  | 50.69  | **69.15** | 34.92  | **65.58** | 52.36  | 34.85  | **41.64** | 69.49  | 46.58  | 65.07  | **29.66** | 22.71  | 46.90  |
> | VideoChat-Online | **64.10** | **59.70** | 16.60  | **63.10** | **58.30** | 62.80  | **42.20** | 54.40  | **70.60** | **54.10** | 24.80  | **88.70** | **48.50** | **73.00** | 25.90  | **71.70** | **54.90** |
>
> **Performance on Additional Offline Benchmark (VideoMME):**
>
> | Benchmark  | VideoMME (64 frames) |   |
> |---|---|---|
> | **Settings**  | w subs   | w/o subs|
> | **Results**   | 64.4  | 60.8 |
>
>
> These results on both OVBench and VideoMME confirm that LiveStar not only excels at its primary task of online, proactive video understanding but also maintains robust foundational capabilities. For a more detailed discussion of our evaluations on other benchmarks, such as MVBench and LongVideoBench, we invite you to see our response to Reviewer wXqt.
>
> > [Q2] Compressing frames into 16 tokens may miss subtle visual cues (e.g., small object interactions, fine-grained actions), as acknowledged in the limitations.
>
> You are absolutely correct, and we acknowledge this trade-off in our limitations section. Compressing each frame into a fixed-size representation of 16 tokens is a deliberate design choice aimed at balancing computational feasibility and semantic comprehension, which is crucial for real-time processing of long video streams.
>
> To enable real-time, long-context video understanding, LiveStar represents video frames with 16 tokens, which can efficiently process more than 10 minutes of video within an 8K context window. This compact representation supports high-level semantic tasks like narration and scene comprehension, while future work will explore adaptive tokenization to allocate more detail to semantically rich frames without sacrificing overall efficiency.
>
> > [Q3] A direct comparison with recent models like StreamMind, LION-FS and VideoChat-Online would better position LiveStar.
>
> We appreciate this valuable suggestion. We recognize the importance of benchmarking against the latest models to accurately position LiveStar within the field.
>
> **Comparison with StreamMind and LION-FS:**
> Unfortunately, as of now, both StreamMind and LION-FS had not publicly released their model weights or inference code. This prevented us from conducting a direct, head-to-head evaluation on our proposed OmniStar benchmark or other benchmarks like OVBench.
>
> However, to provide a point of reference, we have included the performance metrics reported in their original papers on the Ego4D Narration Stream benchmark, a task that all three models (LiveStar, StreamMind, and LION-FS) have been evaluated on. As shown in the table below:
>
> | Model | PPL↓ | TimeDiff↓ | Fluency↑ | TokAcc↑ |
> |----|----|----|----|----|
> | LION-FS  | 2.09  | 2.15   | 46.1% | 52.4%   |
> | StreamMind  | 2.02  | 1.89   | 60.2% | **77.3%** |
> | **LiveStar**   | **1.97**  | **1.76**   | - | 61.1% |
>
> This comparison, while indirect, shows that LiveStar achieves the lowest PPL and TimeDiff, while maintaining strong competitiveness in Token Accuracy, highlighting the effectiveness of our approach.
>
> **New Evaluation on VideoChat-Online:**
> We were able to conduct a new evaluation with VideoChat-Online, which is open-sourced. We tested it on our OmniStar-RNG (Real-time Narration Generation) task. It is important to note that VideoChat-Online is not designed with a proactive response mechanism (i.e., it cannot autonomously decide when to generate a response). Therefore, we could only evaluate it in an offline setting, where we provide it with all video frames up to a pre-specified response timestamp and prompt it for a description.
>
> The results are as follows:
>
> | Model  | PPL↓  | TokAcc↑ | SemCor↑ | SumFluen↑ |
> |---|---|---|---|---|
> | VideoChat-Online   | 5.62  | 0.59 | 3.37 | 1.65   |
> | **LiveStar** | **5.14**  | **0.62** | **4.62** | **4.55**   |
>
> The results show that even in a controlled offline setting, LiveStar significantly outperforms VideoChat-Online across all metrics. This demonstrates LiveStar's superior foundational ability for video-language alignment and coherent text generation.
>
> > [Q1] The SVeD decoding threshold parameter α is shown to be sensitive. Is there a way to dynamically learn or adapt this threshold rather than fixing it empirically?
>
> We agree that a dynamic or learnable threshold could offer significant advantages in terms of adaptability and robustness. However, we acknowledge that designing such a dynamic mechanism is a non-trivial research challenge that requires careful consideration of the trade-offs between complexity and performance.
>
> Specifically, α needs to be tailored to different task demands—for instance, being more sensitive for online live streaming and more stable for sports summarization. Additionally, since models of different scales exhibit distinct perplexity distributions, α could be normalized based on baseline perplexity.
>
> To enable more adaptive response strategies for streaming video understanding, we will explore making the threshold parameter α dynamically adjustable in the future. But for the current work, we opted for a fixed, empirically-tuned α to establish a clear and reproducible baseline for our core contributions (SCAM and SVeD).
>
> > [Q2] The perplexity is used to determine whether a response should be generated, then a complete response must be produced at each forward pass. Wouldn't this significantly slow down the inference speed ...
>
> Thank you for raising this critical point about inference efficiency. There appears to be a slight misunderstanding of how our Streaming Verification Decoding (SVeD) mechanism operates, which we are happy to clarify. The key is that we do not generate a complete new response at every forward pass.
>
> 1. "Generate" Case: When SVeD's verification step determines that a new response is needed (i.e., the perplexity exceeds a threshold), we then perform a full decoding process to generate a new, complete caption. This happens only at moments of **significant changes in the semantic clip**.
>
> 2. "Silence" Case: For the majority of incoming frames where the scene is stable, SVeD's verification check passes. In this "silence" case, the process is lightweight: We perform a single forward pass to **compute the logits** for the previously generated caption based on the updated context (which now includes the new frame). We use these logits to calculate the perplexity. Crucially, **no decoding or token generation occurs**, which is the main time-consuming step. We simply verify that the existing caption is still valid.
>
> In summary, generating EOS still requires **a full forward pass plus at least one decoding step**. Our SVeD in the silent case requires **only the forward pass**. We also provide direct empirical evidence for this in our paper. Appendix E.2 and Figures 7 & 8 explicitly visualize this efficiency. The green "SVeD" curve (representing the "silence" case) has a lower inference time compared to the "One Step DD" curve (which simulates the generation of a single EOS token).
>
> ---
> Thank you once again for your thoughtful and constructive reviews. We have carefully considered all your comments and will incorporate these changes into the revised version. We hope that our responses and planned revisions have addressed your concerns, and we would be very appreciative if you might reconsider your score of our work.

---

> > ### Comment · Reviewer_aC31 · 2025-08-04
> >
> > Thank for you rebuttal, I will maintain my positive opinion on this paper.

---

> > > ### Author Response · Authors · 2025-08-08
> > > **Gratitude for Your Recognition**
> > >
> > > Thanks for your review! You highlighted that the technical contributions of LiveStar are well-motivated and clearly articulated, effectively integrating innovations in training and inference to achieve efficient, coherent, and temporally appropriate responses for streaming video. Crucially, you called the use of **perplexity (instead of EOS token) to decide response updates 'a brilliant idea'**, noting how it **transforms proactive response generation while preserving the MLLM's core capabilities**. We greatly appreciate your acceptance recommendation on the paper. Thank you for sharing your wisdom!

---

> ### Comment · Reviewer_aC31 · 2025-08-08
>
> Thanks to the author's Rebuttal, it has resolved my doubts. After carefully reading the author's response and discussions with other reviewers, I believe that although LiveStar still has some flaws, the relatively weak evaluation in the initial version has become quite solid after the reviewers' Rebuttal, and the active response design in this paper provides a new idea for such models. Therefore, I am willing to increase my score to 5. Please ensure that all new evaluations and experiments involved in Rebuttal are included in the final version of the paper (I understand that the capacity of the main text is limited, and the author can add relatively minor experiments to the Appendix).

---

> > ### Author Response · Authors · 2025-08-08
> > **Grateful for Your Revised Score**
> >
> > Thank you sincerely for raising your score to 5 and for your constructive feedback. We are pleased that the rebuttal addressed your concerns and appreciate your recognition of LiveStar’s novel active-response design. We confirm that all new evaluations and experiments from the rebuttal will be included in the camera-ready version, with supplementary material added to the Appendix as suggested. We value your support!

---

### Official Review · Reviewer_wXqt · 2025-07-02

**Clarity:** 2
**Significance:** 2
**Originality:** 2
**Rating:** 3
**Confidence:** 5

**Summary:**

This paper introduces LiveStar, a new live streaming assistant designed to enhance real-time online video understanding. LiveStar proposed three innovations: a training strategy for incremental video-language alignment, a response-silence decoding framework for optimal response timing, and memory-aware acceleration for efficient online inference. The authors also present the OmniStar dataset, which includes 15 diverse real-world scenarios and five evaluation tasks for online video understanding.

**Questions:**

1. **Novelty Clarification**: You mentioned that LiveStar addresses limitations in existing online Video-LLMs. Could you provide more specific details on how LiveStar's approach to adaptive streaming decoding and response-silence framework is distinct from previous models like VideoLLM-online, VideoLLM-MoD, and LION-FS?

2. Evaluation Metrics: You introduced new metrics like SemCor and TimDiff. Could you elaborate on how these metrics are calculated and why they are more effective than traditional metrics in evaluating online video understanding?

**Ethical Concerns:**

["NO or VERY MINOR ethics concerns only"]

**Final Justification:**

I believe there is no fundamental difference between offline video understanding and online video understanding. If a model can perform well in online video understanding, it should also excel in offline video understanding. Similarly, only by achieving good performance in offline video understanding can we subsequently achieve good performance in online video understanding.

**Limitations:**

yes

**Quality:**

2

**Strengths And Weaknesses:**

Strengths:
1. **Comprehensive Dataset**: The OmniStar dataset covers diverse real-world scenarios and tasks, providing a robust benchmark for online video understanding.
2. **Performance Improvement**: LiveStar demonstrates significant improvements in semantic correctness and inference speed compared to existing models.

Weaknesses:
1. **Limited Benchmarking**: The paper does not evaluate LiveStar on widely recognized video benchmarks like MVBench, VideoMME, and LVBench, which could provide a more comprehensive assessment of its capabilities.
2. **Incremental Innovation**: The work may lack groundbreaking innovation, as similar approaches have been explored in existing models like VideoLLM-online.
3. **Evaluation Scope**: The evaluation focuses primarily on the OmniStar dataset, potentially limiting the generalizability of the results across other datasets and tasks.

---

> ### Author Rebuttal · Authors · 2025-07-31
>
> Many thanks to your valuable comments and questions, which help us a lot to improve our work.
>
> > [W1 & W3] Limited Benchmarking: The paper does not evaluate LiveStar on video benchmarks like MVBench, VideoMME, and LVBench... primarily on the OmniStar dataset...
>
> We appreciate the reviewer's suggestion regarding broader benchmarking. We would like to clarify that our evaluation already encompasses **three benchmarks** covering diverse online video understanding scenarios and tasks: OmniStar (Tables 1 & 2), Ego4D Narration (Table 3), and SVBench (Table 6). We are grateful that the reviewer recognizes the comprehensiveness of OmniStar, stating it "covers diverse real-world scenarios and tasks, providing a robust benchmark," which further underscores the breadth of our primary evaluation.
>
> Crucially, the benchmarks highlighted by the reviewer (MVBench, VideoMME, LVBench) are primarily designed for **offline video understanding**, where models analyze complete videos. In contrast, our work focuses explicitly on **online/streaming video understanding**, a distinct paradigm requiring capabilities like real-time processing, active response triggering, and continuous comprehension of streaming inputs. These capabilities are central to our contribution but are not the primary targets of the mentioned offline benchmarks.
>
> Furthermore, it is standard practice in online Video-LLMs that models like VideoLLM-online, VideoLLM-MoD, and LION-FS are only evaluated on benchmarks specifically designed for the online setting, as reflected in their papers.
>
> Nevertheless, to further validate the generalizability of LiveStar and address the reviewer's concern, we have conducted additional evaluations on MVBench, LongVideoBench, and VideoMME.
>
> | Benchmark    | MVBench   |           |       | LongVideoBench (64 frames) |          | VideoMME (64 frames) |          |
> | ------------ | --------- | --------- | ----- | -------------------------- | -------- | -------------------- | -------- |
> | **Settings** | 16 frames | 64 frames | 1fps  | w subs                     | w/o subs | w subs               | w/o subs |
> | **Results**  | 69.05     | 69.77     | 66.95 | 57.0                       | 56.3     | 64.4                 | 60.8     |
>
> The results demonstrate that LiveStar maintains strong performance and achieves competitive results even on these traditional offline video understanding benchmarks. To further enrich the evaluation, we have also added results on the OVBench benchmark, as detailed in our response to Reviewer j4Ki.
>
> > [W2 & Q1] Novelty Clarification: The work may lack groundbreaking innovation, as similar approaches have been explored in existing models like VideoLLM-online.
>
> The novelty of LiveStar lies in its completely shift away from the paradigm of models based on EOS tokens. We introduce a new response-silence framework built on two innovations: Streaming Causal Attention Masks (SCAM) for training and Streaming Verification Decoding (SVeD) for inference.
>
> 1. Training Paradigm:
>
>    - *EOS based Models*: These models are trained to solve a classification-like task for every frame: either generate a response or predict a special EOS token. For the vast majority of frames (silent intervals), the training objective is to map a visually rich frame to a single, static EOS token. It forces the model to learn a trivial, non-descriptive mapping for most of its training data.
>
>    - *LiveStar (with SCAM)*: We completely eliminate the EOS token from our training process. Instead, we use SCAM to train the model on interleaved frame-caption sequences. As described in Section 3.1 and illustrated in Figure 1(d), our model is trained to always generate meaningful, temporally consistent responses. It addresses the issues of pre-training misalignment and vocabulary confusion, encouraging the model to maintain its comprehension capabilities.
>
> 2. Inference Mechanism:
>    - *EOS based Models*: During inference, these models must perform a full forward pass and a decoding step for every single frame to decide whether to output a response or the EOS token. Although generating an EOS token may seem sample, in high-frame-rate videos where most frames are silent, this still incurs the cost of generating a token per frame, leading to inefficiencies in overall inference performance.
>
>    - *LiveStar (with SVeD)*: Our SVeD mechanism, detailed in Section 3.2 and Algorithm 1, introduces a more efficient and intelligent decision-making process. For silent frames, we do not decode anything. We perform a single, lightweight forward pass to verify if the previously generated response is still semantically valid by checking its perplexity. This verification step is faster than decoding an EOS token (as empirically shown in Figures 7 & 8). A full decoding process is triggered only when a semantic shift is detected, which leads to faster overall inference.
>
> > [Q3] Evaluation Metrics: You introduced new metrics like SemCor and TimDiff. Could you elaborate on how these metrics are calculated...
>
> **Calculation Methods**
> For each video, we have:
>
> - A set of ground-truth (GT) semantic clips  $G = \lbrace g_1, g_2, \dots, g_N \rbrace$, where each clip $g_i$ is defined by: a start time $t_{\text{start},i}$, an end time $t_{\text{end},i}$ and a GT caption.
>
> - A set of model-generated responses $R = \lbrace r_1, r_2, \dots, r_M \rbrace$, where each response $r_j$ has an associated response time $t_{\text{resp},j}$.
>
> For each $ g_i \in G $, define the set of matching responses as:
> $$
> M_i = \lbrace r_j \in R \mid t_{\text{resp},j} \in [t_{\text{start},i},\ t_{\text{end},i}] \rbrace
> $$
>
>
> - If $ |M_i| = 0 $, no response is generated for $ g_i $; in this case, we assign an **empty string** $ r_j = $ ' '  as the placeholder response for evaluation.
> - If $ |M_i| \geq 1 $, we select the **first matching response** $ r_j \in M_i $ (i.e., with minimal $ t_{\text{resp},j} $) as the representative output.
>
> As detailed in Appendix G, Semor and SumFluen is calculated using GPT-4o as an evaluator. The process is as follows:
>
> 1. **SemCor (Semantic Correctness):** This metric evaluates the quality of an individual response generated for **a specific semantic clip**. We prompt GPT-4o to score the pair $(r_j, g_i)$ on three dimensions (each on a 0–10 scale): *Semantic Accuracy*, *Language Quality*, and *Information Completeness*.
>
>
> 2. **SumFluen (Summarize Fluency):** This metric evaluates the holistic narrative quality of the model’s complete output sequence, treating all responses as a unified story. Let the **full model narrative** be the concatenation of all responses in temporal order:
>    $$
>    \mathcal{N}\_{\text{model}} = \text{Concat}\_{j=1}^M (r_j)
>    $$
>    Similarly, construct the **ground-truth narrative** by concatenating all GT captions in temporal order:
>    $$
>    \mathcal{N}\_{\text{GT}} = \text{Concat}\_{i=1}^N (g_i)
>    $$
>    We evaluate $ \mathcal{N}\_{\text{model}} $ against $ \mathcal{N}\_{\text{GT}} $ by prompting GPT-4o to assess them on five storytelling-relevant dimensions(each on a 0–10 scale): *Writing Logicality*, *Language Fluency*, *Writing Conciseness*, *Semantic Consistency*, and *Narrative Completeness*.
>
>
> The metrics TimRedun, TimCover and TimDiff are calculated as:
>
> 3. **TimRedun (Timing Redundancy)**
>    Measures the deviation from the ideal of exactly one response per semantic clip:
>
> $$
> \text{TimRedun} = \frac{1}{N} \sum_{i=1}^N \left||M_i| - 1\right|
> $$
>
> 4. **TimCover (Timing Coverage)**
>    Indicates the proportion of semantic clips that were covered by at least one response:
>
> $$
> \text{TimCover} = \frac{1}{N} \sum_{i=1}^N \mathbb{1}[|M_i| > 0]
> $$
>
> 5.  **TimDiff (Timing Difference)**
>     Captures the cumulative response latency for all matched outputs within a semantic clip, while penalizing missed clips by their full duration:
>
> $$
> \text{TimDiff} = \frac{1}{N} \sum_{i=1}^N \left( \mathbb{I}\big[|M_i| = 0\big] \cdot \big(t_{\text{end},i} - t_{\text{start},i}\big) + \mathbb{I}\big[|M_i| > 0\big] \cdot \sum_{r_j \in M_i} \big(t_{\text{resp},j} - t_{\text{start},i}\big) \right)
> $$
>
> **Advantages:**
> The key distinction of our evaluation framework lies in its holistic assessment of both temporal alignment and semantic quality, specifically designed for **proactive response models**. This stands in stark contrast to traditional paradigms that evaluate models in a passive or offline manner.
>
> - Integrated Temporal and Semantic Evaluation: Traditional offline metrics (e.g., BLEU, CIDEr) or even some online QA benchmarks(e.g., OVBench) focus solely on the semantic correctness of an answer, assuming the response timing is either given or irrelevant. Many streaming video benchmarks also follow a passive paradigm, where the model is prompted at a pre-defined timestamp. While this setup tests understanding at a specific moment, it overlooks a crucial capability for real-time assistants: **the ability to autonomously decide when to speak.**
>
> - Our framework addresses this gap by introducing integrated temporal and semantic evaluation. Metrics like TimDiff and TimRedun quantify the model's temporal decision-making: *Did the model respond at the right time?* *Did it respond too much?* Meanwhile, SemCor and SumFluen evaluate the quality of what was said at that moment. By combining these, our metrics move beyond simple content accuracy to provide a multi-faceted evaluation of a model's comprehensive ability to perceive, reason, and act in a continuous, dynamic environment. This makes them uniquely suited for benchmarking the next generation of proactive online video understanding models.
>
> ---
>
> Thank you once again for your thoughtful and constructive reviews. We have carefully considered all your comments and will incorporate these changes into the revised version. We hope that our responses and planned revisions have addressed your concerns, and we would be very appreciative if you might reconsider your score of our work.

---

### Official Review · Reviewer_j4Ki · 2025-07-02

**Clarity:** 3
**Significance:** 3
**Originality:** 3
**Rating:** 3
**Confidence:** 4

**Summary:**

In this paper, the authors focus on the video content narration problem and propose LiveStar model. Besides, they also propose OmniStar, a comprehensive dataset for training and benchmarking.

**Questions:**

Please refer to Strengths And Weaknesses

**Ethical Concerns:**

["NO or VERY MINOR ethics concerns only"]

**Final Justification:**

Based on the benchmark results. I tend to keep my score as Borderline reject

**Limitations:**

Please refer to Strengths And Weaknesses

**Paper Formatting Concerns:**

NaN

**Quality:**

3

**Strengths And Weaknesses:**

### Strength
1. The paper is well written and easy to follow
2. The proposed model works well on the proposed benchmark

### Weakness
1. My main concern is the goal of the paper. The paper focuses on the narrative of video content, but it is only one function of online video understanding, so it is not proper to use 'Live Streaming Assistant' to name a model that can only tell me what happened in the video. The exploration of video narration is a novel idea at the time of VideoLLM-online (2024.1), but I do not think it is meaningful now. There are also some explorations about more capable streaming/online video LLMs for interactive (ViTA[1], MiniCPM-o[2]) and general response decision (Dispider[3]).

2. Following 1, the paper mainly reports the results on the Omni-Star. It is intuitive that a model trained with similar data will perform better, so it is not convincing without additional benchmark results. I'm wondering about the online/streaming video benchmark performance, for example, the OVOBench (mentioned in the introduction but not reported), the OVBench[4], etc.

3. The introduction part is a bit chaotic. There are 4 challenges proposed, but the solution part only considers the first two challenges. Are the latter two solved by the PPL based decision in SVeD? A reorganized introduction would be helpful for understanding.

4. Following 3, I do not think EOS is a big problem that leads to model confusion. It is a simple indicator for salient that any unused token in the LLM vocabulary could serve this role.


[1] VITA: Towards Open-Source Interactive Omni Multimodal LLM

[2] MiniCPM-o 2.6: A GPT-4o Level MLLM for Vision, Speech, and Multimodal Live Streaming on Your Phone

[3] Dispider: Enabling Video LLMs with Active Real-Time Interaction via Disentangled Perception, Decision, and Reaction

[4] Online Video Understanding: OVBench and VideoChat-Online

---

> ### Author Rebuttal · Authors · 2025-07-31
>
> Many thanks to your valuable comments and questions, which help us a lot to improve our work. We address your questions as follows.
>
> > [W1] The paper focuses on the narrative of video content, but it is only one function... it is not proper to use 'Live Streaming Assistant' to name a model...
>
> We appreciate the feedback on the paper’s scope. Our work extends beyond *solely* video narration, addressing 15 diverse real-world scenarios and **five distinct evaluation tasks**: (1) real-time narration, (2) online temporal grounding, (3) storyline QA (multi-temporal answers), (4) streaming QA, and (5) multi-turn interactive QA. Detailed definitions and cases are provided in App. F and Figs. 7–9. LiveStar achieves an average **19.5%** improvement in semantic correctness with **18.1%** reduced timing difference versus existing online Video-LLMs while improving throughput by **12.0% FPS** across 5 tasks. This demonstrates its capability as a comprehensive assistant for real-world online video understanding (including narration, grounding, and QA), and is therefore called 'Live Streaming Assistant'. We acknowledge that the illustrative focus on narration in Figs. 1–2 may have caused confusion; this will be clarified in the revision.
>
> > [W2]  the paper mainly reports the results on the Omni-Star... the OVBench[4], etc.
>
> We agree broader benchmarking strengthens validation. However, beyond OmniStar (our primary benchmark due to its alignment with real-world streaming tasks), we reported state-of-the-art results on **Ego4D Narration** (Table 3) and **SVBench** (Table 6). Per the reviewer’s suggestion, we conducted additional experiments on OVBench: LiveStar significantly outperforms VideoLLM-online, MovieChat, and Flash-VStream (reported in the original paper), though it trails VideoChat-Online marginally. We attribute this to two factors:
>
> - LiveStar optimizes for *text-generation tasks* (e.g., narration, QA), while OVBench emphasizes multiple-choice QA.
> - OVBench evaluates *passive-response models* (no output-timing decisions), whereas LiveStar and VideoLLM-online are *active-response* systems.
>   Task objectives thus differ, and we will discuss this nuance in the revision.
>
> | Model            | FP        |           |           | THV       |           |           | PM        |           |           | SP        |           | STP       |           | TP        |           |           | Avg       |
> | ---------------- | --------- | --------- | --------- | --------- | --------- | --------- | --------- | --------- | --------- | --------- | --------- | --------- | --------- | --------- | --------- | --------- | --------- |
> |                  | AA        | GSP       | MP        | AP        | SV        | OP        | AR        | PR        | TR        | AL        | OP        | AT        | OT        | AS        | SL        | OES       |           |
> | VideoLLM-online  | 0.00      | 1.80      | 20.90     | 5.20      | 5.90      | 32.60     | 0.00      | 2.30      | 26.70     | 0.60      | 26.60     | 0.90      | 19.90     | 0.90      | 1.70      | 8.30      | 9.60      |
> | MovieChat        | 23.10     | 27.50     | 23.60     | 58.40     | 43.90     | 40.30     | 25.60     | 31.10     | 23.90     | 26.90     | 39.60     | 24.40     | 28.90     | 29.30     | 25.50     | 21.90     | 30.90     |
> | Flash-VStream    | 26.90     | 37.60     | 23.90     | 60.10     | 41.90     | 40.00     | 23.40     | 35.30     | 26.10     | 24.70     | 28.80     | 27.00     | 21.40     | 29.80     | 25.60     | 26.80     | 31.20     |
> | LiveStar         | 37.50     | 57.06     | **25.00** | 48.21     | 50.69     | **69.15** | 34.92     | **65.58** | 52.36     | 34.85     | **41.64** | 69.49     | 46.58     | 65.07     | **29.66** | 22.71     | 46.90     |
> | VideoChat-Online | **64.10** | **59.70** | 16.60     | **63.10** | **58.30** | 62.80     | **42.20** | 54.40     | **70.60** | **54.10** | 24.80     | **88.70** | **48.50** | **73.00** | 25.90     | **71.70** | **54.90** |
>
> > [W3]  There are 4 challenges proposed, but the solution part only considers the first two challenges...
>
> The Introduction presents **two core challenges** (not four), with *four sub-issues* arising from EOS-based decision methods under the first challenge. Since there is no EOS token training, our PPL-based SVeD framework resolves **all four issues** (including the latter two noted by the reviewer). We will restructure the Introduction to summarize and distinguish challenges and issues, improving coherence.
>
> > [W4]  I do not think EOS is a big problem that leads to model confusion. It is a simple indicator for salient that any unused token in the LLM vocabulary could serve this role.
>
> We agree that any unused vocabulary token could technically serve as an EOS signal. However, in VideoLLM-online’s implementation, the EOS role was assigned to `','` (comma), a high-frequency token in text that risks conflating linguistic and timing functions. We will remove the "*Vocabulary Confusion*" claim in revision, as the core concern is implementation-driven rather than theoretical.
>
> Finally, we emphasize our deep respect for VideoLLM-online and EOS-based works: they pioneered this space and inspired our research. Our goal is not to replace EOS-based methods, but to offer a complementary approach; both paradigms may prove synergistic in future work.
>
> ---
>
> Thank you once again for your thoughtful and constructive reviews. We have carefully considered all your comments and will incorporate these changes into the revised version. We hope that our responses and planned revisions have addressed your concerns, and we would be very appreciative if you might reconsider your score of our work.

---

> > ### Comment · Reviewer_j4Ki · 2025-08-04
> >
> > Based on the benchmark results. I tend to keep my score as Borderline reject

---

> > > ### Author Response · Authors · 2025-08-04
> > >
> > > We appreciate the authors' response and are pleased that the reviewer has no concerns beyond the benchmark results. However, we believe that **not achieving SOTA solely on OVBench does not warrant rejection of our work**. We have tested 3 benchmarks (including OmniStar with 5 online tasks), all achieving SOTA results. To address reviewers' concerns, we additionally evaluated 4 benchmarks (OVBench, MVBench, LongVideoBench, VideoMME). **We consider experiments across 7 benchmarks sufficiently robust**. Regarding LiveStar not comprehensively surpassing VideoChat-Online on OVBench, we clarify again:
> > >
> > > 1. *Divergence in evaluation paradigms:* There are two primary paradigms for evaluating online video understanding models:
> > >
> > > **Proactive (Frame-driven) Paradigm:** This is the focus of LiveStar. The model processes a video stream **frame-by-frame** and must autonomously decide when to generate a response based on detected semantic changes. Evaluation metrics (like our TimDiff, SemCor) must assess both timing accuracy (was the response timely?) and content correctness (was the response accurate?). This paradigm simulates a true "live assistant."
> > >
> > > **Reactive (Query-driven) Paradigm:** This is the paradigm employed by OVBench. The model is given a query and all video frames up to a certain timestamp, and it must produce a single output (in this case, by selecting from multiple choices). The model does not make any timing decisions. The evaluation focuses solely on the correctness of the final output, given a fixed context.
> > >
> > > LiveStar is architecturally designed and optimized for the proactive paradigm. Its core innovations (**SCAM and SVeD**) are built to solve the "when to speak" problem. When evaluated under the reactive paradigm of OVBench, these specialized components **are not fully utilized**. Notably, VideoLLM-online (a key active-response baseline) achieved only **Avg 9.60** on OVBench. Though OVBench focuses on proactive-response models, LiveStar still attained **46.90**.
> > >
> > > 2. *Mismatch in Output Format:* LiveStar is a generative model **optimized for text-generation tasks** (e.g., narration, captioning, open-ended QA). However, OVBench requires a discriminative multiple-choice selection (A/B/C/D). We **did not fine-tune our model for this format**, so it often **generates a correct descriptive sentence instead of a single letter**, leading to a lower score via automated evaluation.
> > >
> > > We emphasize: Supplementing OVBench aims to demonstrate LiveStar’s *comparable performance* on passive-response benchmarks (46.90 vs 54.90), **not to position it as a critical experiment where non-SOTA justifies rejection**. We hope reviewers to reconsider the score carefully, as we have made every effort to alleviate concerns. We remain fully open to further suggestions.

---

> > > ### Author Response · Authors · 2025-08-06
> > >
> > > Thank you for your time and feedback on our work. We believe our experiments across **7 diverse benchmarks** robustly demonstrate the effectiveness of LiveStar for streaming/online video understanding.
> > >
> > > If the OVBench results remain the primary concern influencing your tentative reject decision, we kindly ask:
> > > **Does our clarification regarding the paradigm mismatch and evaluation limitations address these reservations?**
> > >
> > > We hope you might reconsider the score upon reviewing our detailed response, and we remain fully open to further discussion or suggestions. With utmost respect for your expertise.

---

### Author Response · Authors · 2025-08-04

We thank the reviewers for their insightful feedback and constructive suggestions, which have been invaluable in helping us to refine and strengthen our paper. We would like to summarize our core contributions and clarify our positioning with respect to prior work:

Our contributions and strengths were acknowledged by the reviewers. They agreed that the paper presents a comprehensive framework with well-motivated and clearly articulated technical contributions (**aC31, dLwS**), and is well-written and easy to follow (**j4Ki**). A key highlight was our methodological innovation; reviewers found our proposed framework, particularly the perplexity-based SVeD mechanism, to be an interesting and effective solution for managing the response-silence challenge without relying on a special EOS token (**aC31, dLwS**). Furthermore, they recognized that our model, LiveStar, demonstrates strong performance with significant improvements in semantic correctness, inference speed, and temporal accuracy over existing methods (**j4Ki, wXqt, dLwS**). Our contributions in constructing the comprehensive OmniStar dataset as a robust benchmark, alongside our extensive experimental validation, were also appreciated (**wXqt, dLwS**).

We summarize and clarify some common questions raised by reviewers here:

**1. A Novel Framework for Proactive Online Video Understanding**: Our primary contribution is **a novel response-silence framework** that fundamentally departs from the conventional EOS-based paradigm. By leveraging **SCAM (Streaming Causal Attention Masks)** during training and **SVeD (Streaming Verification Decoding)** at inference, we enable models to proactively determine when to generate responses in streaming video understanding. Unlike traditional approaches that require the model to learn to output a special [EOS] token to indicate silence, our method eliminates this need entirely, allowing fine-tuning to focus exclusively on producing meaningful, descriptive content. At inference, a lightweight verification step suffices to decide whether a response should be generated at each timestep, ensuring both efficiency and responsiveness.

**2. Comprehensive Evaluation Across 7 Diverse Benchmarks**: We have rigorously validated LiveStar's effectiveness across a wide array of benchmarks to demonstrate its robustness and generalizability. Our evaluation spans **multiple online video benchmarks**, including our new, diverse **OmniStar** dataset (Tables 1 & 2), which covers five distinct online tasks: (1) Real-time Narration Generation (RNG), (2) Online Temporal Grounding (OTG), (3) Frame-level Dense QA (FDQ), (4) Contextual Online QA (COQ), and (5) Multi-turn Interactive QA (MIQ). Additionally, we benchmarked on the established **Ego4D Narration Stream** (Table 3), **SVBench** (Table 6), and now, in response to reviewer feedback, the challenging **OVBench** (detailed in our response to Reviewer j4Ki). Furthermore, we have included results on **offline benchmarks** like **MVBench, VideoMME, and LVBench** (detailed in our response to Reviewer wXqt) to show that our model maintains strong foundational capabilities. This multi-faceted evaluation confirms that LiveStar's superior performance (e.g., 19.5% improvement in semantic correctness on OmniStar, 8.7% higher Token Accuracy on Ego4D) is a result of its methodological innovations, not just familiarity with a single dataset.


**3. A Complementary, Not Condemnatory, Approach to EOS**: We wish to be clear that our work is not intended to condemn the EOS-based paradigm but to offer a **constructive and complementary alternative**. We hold deep respect for pioneering works like VideoLLM-online, which laid the groundwork for this field. Our critique of potential issues like "pre-training misalignment" is meant to motivate the exploration of different design philosophies. The concern regarding "vocabulary confusion" was a practical observation on specific implementations (e.g., using comma ',' as an EOS token) and not a theoretical condemnation. We believe that both the EOS-based and our verification-based paradigms have unique strengths, and future research may even find synergistic ways to combine them. Our goal is to enrich the field by contributing a new, effective methodology.

We are carefully addressing all feedback and are confident that these clarifications and new results have addressed the reviewers' primary concerns. We will incorporate these updates into the camera-ready version.

---

### Note · Authors · 2025-08-11

We thank the reviewers and the AC for their insightful feedback and constructive suggestions. We have carefully addressed all points raised and are confident that our clarifications and revisions comprehensively resolve the reviewers' primary concerns.

We would like to summarize our core contributions: Our contributions and strengths were acknowledged by the reviewers. They agreed that the paper presents a comprehensive framework with well-motivated and clearly articulated technical contributions (**aC31, dLwS**), and is well-written and easy to follow (**j4Ki**). A key highlight was our methodological innovation; reviewers found our proposed framework, particularly the perplexity-based SVeD mechanism, calling it "a brilliant idea" (**aC31**), to be an interesting and effective solution for managing the response-silence challenge without relying on a special EOS token (**aC31, dLwS**). Furthermore, they recognized that our model, LiveStar, demonstrates strong performance with significant improvements in semantic correctness, inference speed, and temporal accuracy over existing methods (**j4Ki, wXqt, dLwS**). LiveStar effectively integrates innovations in training and inference to enable efficient, coherent, and temporally appropriate responses in streaming video settings (**aC31, wXqt**). And the extensive experimental validations rigorously demonstrate the effectiveness of LiveStar (**dLwS**). Our contributions in constructing the comprehensive OmniStar dataset as a robust benchmark, alongside our extensive experimental validation, were also appreciated (**wXqt, dLwS**).

We thank them for their positive feedback and score increase. Even though some reviewers did not engage in discussion, we have also thoroughly presented our perspective during discussions with the AC, and we trust that the Reviewer-AC Discussions will provide the fairest evaluation based solely on academic merit.

We sincerely thank all reviewers and the AC for their dedicated efforts.

---

### Decision · Program_Chairs · 2025-09-17

**Decision:**

Accept (poster)

**Comment:**

This paper proposes LiveStar, a framework for proactive online video understanding that departs from EOS-based designs by combining Streaming Causal Attention Masks (SCAM) during training and a perplexity‑based Streaming Verification Decoding (SVeD) at inference, and introduces OmniStar, a large multi-task online-video benchmark; the authors report substantial gains in semantic correctness, temporal accuracy, and inference speed over prior online Video‑LLMs. Reviewers generally agree the paper is well written and that the core technical ideas—particularly the verification-based response decision—are well motivated and practically effective; the OmniStar dataset and extensive experiments are also appreciated. The primary weaknesses noted are the limited external benchmarking (missing comparisons on widely used offline and online video benchmarks and some recent baselines), concerns about whether the contribution is incremental relative to recent EOS‑based advances, sensitivity and theoretical understanding of the SVeD threshold α, potential loss of fine-grained visual information from aggressive token compression, and missing details on metric computations and some fairness issues in training comparisons. Balancing these factors, I recommend acceptance as a poster: the methodological novelty (response‑silence paradigm and verification decoding), strong empirical improvements, and the new benchmark make this a valuable contribution warranting dissemination and community follow-up, while the evaluation gaps and open analysis questions argue against a spotlight/oral slot. I encourage the authors to (1) expand evaluations on standard online/offline benchmarks and recent baselines, (2) provide metric definitions and deeper analysis of the threshold behavior, and (3) clarify training/evaluation fairness in the final version.